## RESEARCH ARTICLE

# The RNA-binding protein SRSF3 controls epicardial formation by regulating splicing and proliferation

Irina-Elena Lupu[1],[*], Susann Bruche[1],[*], Anob M. Chakrabarti[2],[3], Ian R. McCracken[1], Quang M. Dang[1],[4], Tamara Carsana[1], Sarah De Val[1], Andia N. Redpath[1],[‡] and Nicola Smart[1],[‡],[§]

## ABSTRACT

The epicardium is a fundamental regulator of cardiac development and regeneration, functioning to secrete essential growth factors and to produce epicardium-derived cells that contribute coronary mural cells and cardiac fibroblasts. The molecular mechanisms controlling epicardial formation have not been fully elucidated. In this study, we report that the RNA-binding protein SRSF3 is highly expressed in the embryonic proepicardium and epicardial layer. Deletion of *Srsf3* from the murine proepicardium led to proliferative arrest, preventing proper epicardial formation. Induction of *Srsf3* deletion after the proepicardial stage resulted in impaired epicardial proliferation and epicardium-derived cell formation. Single-cell RNA sequencing showed that SRSF3-depleted epicardial cells were eliminated; however, the surviving non-recombined cells upregulated *Srsf3*, became hyperproliferative and, remarkably, compensated for the early deficit. This unexpected finding attests to the importance of SRSF3 in controlling epicardial proliferation, and highlights the significant confounding effect of mosaic recombination on embryonic phenotyping. Mapping the SRSF3–RNA interaction network by endogenous irCLIP identified binding to major cell cycle regulators, including *Ccnd1* and *Map4k4*, mediating both splicing and non-splicing roles. This research defines SRSF3 as an important regulator of epicardial formation and function.

KEY WORDS: Epicardium, Proliferation, SRSF3, Mosaic recombination, Inducible Cre recombinase, Tamoxifen, Mouse

## INTRODUCTION

Understanding cardiac morphogenesis is essential for designing improved therapies to treat heart disease. The outer layer of the heart, the epicardium, plays a major role in cardiac formation and repair; it has been shown to promote cardiomyocyte proliferation and vessel growth in developing embryos and in species that regenerate their hearts, through provision of growth factors and perivascular support (Cao et al., 2016; Cao and Cao, 2018; Dubé et al., 2017; Kikuchi et al., 2011; Lepilina et al., 2006; Redpath and Smart, 2021; Smart and Riley, 2012; Wang et al., 2015). For non-regenerative species, maximising the therapeutic potential of the epicardium depends upon augmenting the restricted reactivation and proliferation that occur endogenously (Redpath and Smart, 2021). The epicardial layer originates from the proepicardial organ (PEO), a transient embryonic structure located at the venous pole of the developing heart (Schulte et al., 2007). After epicardial formation, around embryonic day (E) 11.5 in mouse, some epicardial cells undergo epithelial-to-mesenchymal transition (EMT), leading to the formation of epicardium-derived cells (EPDCs) that contribute most of the vascular smooth muscle cells of the heart and cardiac fibroblasts (Liu et al., 2016; Mellgren et al., 2008; Rudat et al., 2013; Singh et al., 2016; Smith et al., 2011).

A key process in epicardial formation and function is cell proliferation. PEO cluster emergence in zebrafish has been shown to be dependent on tension generated through regionalised proliferation of mesodermal progenitors (Andrés-Delgado et al., 2019). Epicardial EMT also appears to require cell division, with orientation of the mitotic spindle dictating whether cells remain on the surface or invade the myocardium (Wu et al., 2010). Studies in which epicardial proliferation rate was increased by cyclin D1 overexpression (Wu et al., 2010) or deletion of neurofibromin 1 (NF1), a negative regulator of Ras (Baek and Tallquist, 2012), resulted in increased formation of EPDCs.

The RNA-binding protein serine/arginine-rich splicing factor 3 (SRSF3) is the smallest member of the serine/arginine rich (SR) family of proteins, master regulators of RNA metabolism, involved in processes such as mRNA transcription, splicing, export, stability and translation (Änkö et al., 2012; Kim et al., 2014; Müller-McNicoll et al., 2016; Mure et al., 2018; Park and Jeong, 2016; Ratnadiwakara et al., 2018; Shen et al., 2004; Shepard and Hertel, 2009; Zahler et al., 1992). SRSF3 is characterised as an oncogene due to its role in promoting cell proliferation and migration in various cancers (Corbo et al., 2013; Kano et al., 2013; Ke et al., 2018; Kim et al., 2017; Kurokawa et al., 2013; Park and Jeong, 2016; Sen et al., 2015; Tang et al., 2013). SRSF3 has been shown to be required for cardiomyocyte proliferation during development, and to prevent de-capping of contraction-related mRNA transcripts, with conditional deletion of SRSF3 from this lineage resulting in embryonic lethality mid-gestation (Ortiz-Sánchez et al., 2019). SRSF3 has been implicated in multiple developmental transitions (Änkö et al., 2010; Do et al., 2018; Jumaa et al., 1999; Sen et al., 2013), but the role of SRSF3 in the epicardium has not been previously investigated.

Here, we show that SRSF3 is expressed ubiquitously in the heart during embryonic development, with highest expression in the PEO

[1]Institute of Developmental and Regenerative Medicine, British Heart Foundation Centre of Research Excellence, Department of Physiology, Anatomy and Genetics, University of Oxford, Oxford OX3 7TY, UK. [2]RNA Virus Replication Laboratory, The Francis Crick Institute, London, NW1 1AT, UK. [3]UCL Respiratory, Division of Medicine, University College London, London, WC1E 6BT, UK. [4]Department of Medical Genetics and Rare Diseases, Vinmec Healthcare System, Hanoi, Vietnam.
*These authors contributed equally to this work
[‡]These authors contributed equally to this work

[§]Author for correspondence (nicola.smart@dpag.ox.ac.uk)

I.-E.L., 0000-0002-0869-8525; S.B., 0000-0002-5814-7166; A.M.C., 0000-0002-6841-5718; I.R.M., 0000-0001-8806-548X; Q.M.D., 0009-0000-7083-8560; T.C., 0009-0004-3387-3171; S.D.V., 0000-0002-2566-2348; A.N.R., 0000-0002-3620-4983; N.S., 0000-0002-3501-7254

and epicardium until E12.5. To address the role of SRSF3 in the epicardium, we generated two SRSF3 conditional knockout mice. Deletion of SRSF3 using the Tg(Gata5-Cre) line resulted in impaired epicardial layer formation, due to decreased proliferation of founder cells in the PEO, and embryonic lethality at E12.5. Epicardial-specific deletion later in development, using the inducible Wt1[CreERT2] line, resulted in a less-severe phenotype characterised by impaired coronary vasculature formation, reduced cardiac compaction and myocardial hypoxia, due to defective epicardial proliferation and EMT. However, mosaic recombination allowed non-targeted epicardial cells to hyperproliferate, paradoxically upregulating *Srsf3* to drive a robust compensatory mechanism and virtually rescue the phenotype by late embryonic stages. To delineate the molecular mechanisms through which SRSF3 controls epicardial proliferation, we used single-cell and bulk RNA sequencing, to define the transcriptional changes that result from epicardial *Srsf3* depletion. We also employed individual nucleotide resolution infrared cross-linking and immunoprecipitation (irCLIP) to map SRSF3-RNA binding sites across the epicardial transcriptome. These analyses confirmed control of mitotic cell cycle as a primary function of SRSF3 in the epicardium and demonstrated direct binding to transcripts encoding key regulators of proliferation, such as cyclin D1, and senescence, including MAP4K4. These data reveal an essential role for SRSF3 in the proliferation of (pro)epicardial cells, and in the epicardial processes that underpin cardiac morphogenesis.

## RESULTS

### Characterisation of SRSF3 expression during development

The expression pattern of SRSF3 at key stages of epicardial activity was investigated by immunohistochemistry (IHC) of mouse embryo and heart cryosections, including at E9.5, when epicardial progenitors first arise within the PEO; at E11.5, when the epicardium is most proliferative and has formed a layer (Lupu et al., 2020b; Wu et al., 2010); and at E15.5 when the epicardium downregulates key marker genes, such as *Wt1* (Lupu et al., 2020b), which encodes a transcriptional regulator synonymous with the active epicardial state (Redpath and Smart, 2021), and progresses towards quiescence (Liu et al., 2016; Wu et al., 2010). The highest levels of SRSF3 were found in the PEO and in the early epicardium (E11.5), after which SRSF3 expression started to decline (Fig. 1A, B, Fig. S1A), remaining low postnatally (Fig. S1B). Notably, the reduction in SRSF3 paralleled the previously characterised downregulation of Wilms' tumor 1 (WT1) in the epicardium (Lupu et al., 2020b). Strong SRSF3 expression was observed in E11.5 epicardial explant cultures (Fig. 1C), in which outgrowth is governed by epicardial cell proliferation.

### SRSF3 is required for epicardial formation and cardiomyocyte survival

To investigate the role of SRSF3 in the PEO, Tg(Gata5-Cre); Rosa26[TdTomato] mice (Madisen et al., 2010; Merki et al., 2005) were crossed with *Srsf3[fl/fl]* mice (Ortiz-Sánchez et al., 2019) in order to deplete SRSF3 in epicardial progenitor cells and concomitantly label them to track their migration and fate (defined as *Srsf3* cKO; Fig. 1D). Tg(Gata5-Cre) uses a chick enhancer of *Gata5* that becomes active from E9.25 and drives Cre recombinase expression in the PEO (Merki et al., 2005), septum transversum and a subset of cardiomyocytes (Vieira et al., 2017). In line with previous reports (Carmona et al., 2020; Sedmera et al., 2024), we observed Cre-based reporter labelling of >50% of cardiomyocytes, with a patchy distribution across both right and left ventricles (Figs S1, S2). Despite the septum transversum and cardiomyocyte recombination,

Tg(Gata5-Cre) enables efficient targeting of E9.5 proepicardial progenitors, which cannot be achieved with inducible epicardial Cre lines. Reduced SRSF3 expression was demonstrated in E11.5 cultured epicardial explants (Fig. S1C); TdTomato-positive cells generally presented the lowest levels of SRSF3, which was notably absent from nuclear speckles, where splicing factors normally reside (Tripathi et al., 2010). However, the correlation between tdTomato expression and SRSF3 depletion was imperfect; as widely reported (Fernandez-Chacon et al., 2019; Payne et al., 2018; Redpath and Smart, 2021), the highly efficient recombination of the short transcriptional stop sequence of Cre reporters does not accurately reflect recombination across multiple exons, meaning that gene deletion is typically much less efficient.

*Srsf3* cKO displayed embryonic lethality, with no live embryos recovered beyond E12.5 (Table S1). E12.5 *Srsf3* cKO embryos presented gross morphological abnormalities of the heart, including hypoplastic ventricles and dilated atria (Fig. 1E). To investigate the impact of SRSF3 depletion on epicardial formation, immunostaining for WT1 was performed on E12.5 heart sections (Fig. 1F, Fig. S1D). Fewer WT1-positive cells were detected on the surface of *Srsf3* cKO hearts compared to controls (Fig. 1F,G). It is important to note that ~90% of epicardial cells were tdTomato-labelled in E12.5 control hearts, as Tg(Gata5-Cre) was not active in all epicardial progenitor cells. The proportion of epicardial cells positive for tdTomato was significantly decreased in *Srsf3* cKO compared to controls (Fig. 1H, Fig. S1D), suggesting that non-targeted, tdTomato-negative cells are disproportionately either more proliferative or more likely to translocate onto the heart, compared with those in which *Srsf3* was successfully depleted. Since the ventricles were smaller, cell death and proliferation were investigated by IHC to reveal increased cell death and decreased proliferation in *Srsf3* cKO hearts, as indicated by increased cleaved-caspase 3 (CC3) and decreased phospho-histone H3 (PHH3), respectively (Fig. S1E-H). All cells expressing CC3 were positive for sarcomeric α-actinin (s-α-actinin), suggesting that SRSF3 is only required for survival of cardiomyocytes (Fig. S1E). There were fewer PHH3/tdTomato-positive cardiomyocytes and epicardial cells in *Srsf3* cKO hearts, compared with control hearts (Fig. S1G,H), suggesting that SRSF3 is needed for the proliferation of both cell types. The requirement for SRSF3 to enable cardiomyocyte proliferation in the developing embryo has been reported (Ortiz-Sánchez et al., 2019); however, the involvement of SRSF3 in epicardial proliferation has not been addressed.

### SRSF3 depletion in the PEO results in impaired proliferation and migration of epicardial progenitor cells

The impaired epicardial formation phenotype was more closely investigated in ventricular epicardial explants from E11.5 *Srsf3* cKO embryos, which demonstrated reduced outgrowth (Fig. 2A,B) and a lower proportion of tdTomato positive cells compared with control (Fig. 2C). To assess whether epicardial formation is disrupted from the outset, we checked for the presence of epicardial cells on the surface of the heart at E10.5, when proepicardial progenitor cells complete their translocation onto the heart. Few WT1-positive epicardial cells were detected on E10.5 *Srsf3* cKO hearts, in contrast to the extensive coverage attained in controls (Fig. 2D, Fig. S2A). This prompted us to determine whether the formation of the proepicardium itself was impaired by loss of *Srsf3*. Indeed, we observed fewer epicardial progenitor cells in the PEO of *Srsf3* cKO embryos at E9.5 compared to controls, with mostly non-targeted, tdTomato-negative cells present (Fig. 2E, white arrows). This was likely a consequence of impaired proliferation, as demonstrated by diminished Ki67 expression (Fig. S2B). To confirm the

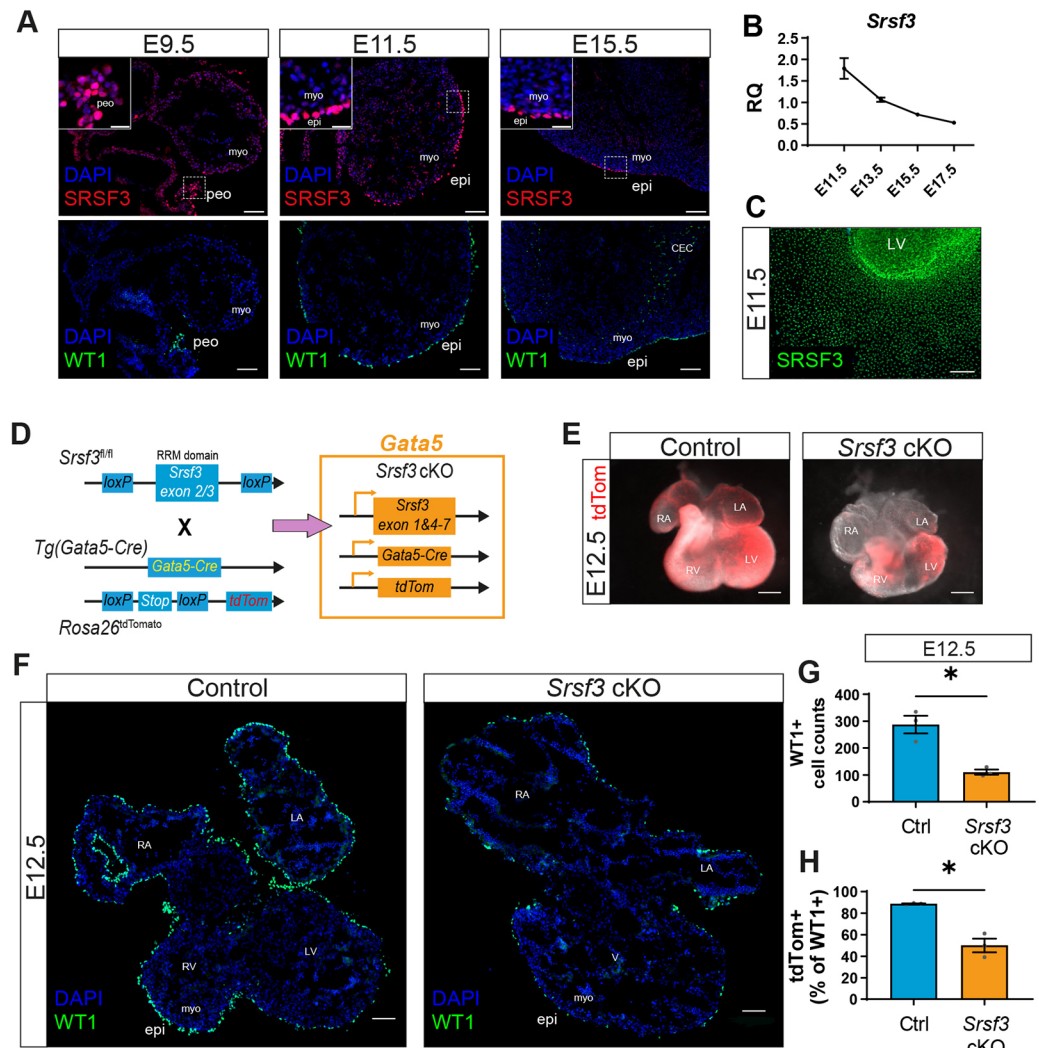

**Fig. 1. SRSF3 is expressed in the developing heart and is required for epicardium formation.** (A) Immunofluorescence of SRSF3 and WT1 in embryonic mouse hearts at E9.5, E11.5 and E15.5. Images representative of *n*>4 embryos. Insets show magnifications of the boxed areas. (B) qRT-PCR analysis of *Srsf3* transcript in whole heart lysates at E11.5, E13.5, E15.5 and E17.5. RQ, relative quantification. Values normalised to *Actb*. Error bars indicate mean±s.e.m. (*n*=5). (C) Immunofluorescence of SRSF3 in an E11.5 epicardial explant. Image representative of *n*=8 embryos. (D) Generation of a conditional SRSF3 knockout mouse (*Srsf3* cKO) with lineage tracing capacity, targeting the epicardial lineage [Tg(Gata5-Cre)] and reported by tdTomato (Rosa26^tdTomato). (E) Fluorescence stereomicroscope images of control and *Srsf3* cKO embryonic hearts at E12.5. Epicardial lineage reported by tdTomato fluorescence. Images representative of *n*=3 embryos. (F) Immunofluorescence showing WT1 in control and *Srsf3* cKO embryonic hearts at E12.5. Images representative of *n*=4 embryos. (G,H) Quantification of WT1$^+$ cells (G) and the percentage tdTomato+ epicardial lineage cells as a proportion of total WT1$^+$ cells (H). Error bars indicate mean±s.e.m. (*n*=3). *P<0.05 (unpaired *t*-test with Welch's correction). CEC, coronary endothelial cell; epi, epicardium; LA, left atrium; LV, left ventricle; myo, myocardium; peo, proepicardium; RA, right atrium; RV, right ventricle; V, ventricles. Scale bars: 100 μm (A,F); 200 μm (C); 500 μm (E).

proliferation defect, and more accurately quantify progenitor expansion, PEO explants were cultured from control and *Srsf3* cKO embryos (Fig. 2F-H, Fig. S2C). As expected, *Srsf3*-depleted PEO explants were significantly smaller than controls (mean 1166 versus 3654 cells per explant; Fig. 2F, Fig. S2C) and contained 56% fewer Ki67-positive cells (Fig. 2G,H), further supporting the notion that SRSF3 is required for proliferation of epicardial precursors.

### SRSF3 depletion in the epicardium leads to impaired proliferation and enhanced cell death in epicardial cells

To bypass the proepicardial defects and investigate SRSF3 function exclusively in the epicardium, we used the inducible Wt1^CreERT2 line (Xiao et al., 2018; Zhou et al., 2008), with tamoxifen induction limited to E9.5-E11.5 to avoid significant targeting of coronary

endothelial cells (Lupu et al., 2020b). This strategy was adopted to generate an inducible *Srsf3* KO (*Srsf3* iKO) with capacity to trace and temporally target the epicardial lineage (Fig. 3A). The restricted window for induction prioritises specificity over efficiency. While coronary endothelial cell and cardiomyocyte targeting can be largely avoided with this strategy, it produces mosaic recombination, even with our optimised tamoxifen regime. Thus, while both constitutive and inducible epicardial Cre lines are imperfect, they are considered the best available, and their combined use partly mitigates their respective inadequacies.

Embryonic hearts were initially evaluated at E13.5 when the epicardium is fully formed and EMT is underway. Despite having a largely intact epicardium, *Srsf3* iKO hearts exhibited a disrupted morphology and markedly fewer lineage-traced EPDCs invading

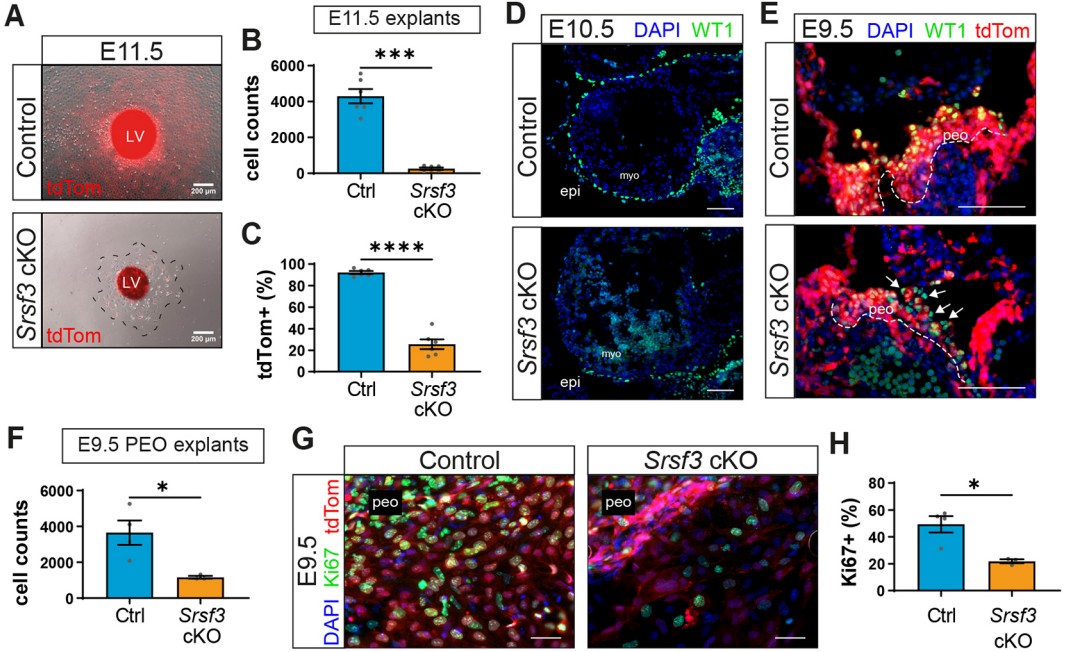

**Fig. 2. Impaired proliferation of epicardial progenitor cells in *Srsf3* cKO hearts.** (A) Brightfield and tdTomato fluorescence images of control and *Srsf3* cKO E11.5 ventricular explants (3 days' culture). Dashed line delineates the extent of epicardial outgrowth. (B,C) Quantification of epicardial cells (B) and percentage of tdTomato+ cells (C). Error bars indicate mean±s.e.m. (*n*=6). ***P<0.001, ****P<0.0001 (unpaired *t*-test with Welch's correction). (D) Immunofluorescence images showing WT1+ cells on the surface of control and *Srsf3* cKO embryonic mouse hearts at E10.5. Images representative of *n*=3 embryos. (E) Immunofluorescence showing WT1+ epicardial progenitor cells situated in the PEO (outlined by dashed line) of control and *Srsf3* cKO embryos at E9.5. White arrows highlight non-targeted cells. Images representative of *n*=5 embryos. (F) Quantification of (pro)epicardial cells from control and *Srsf3* cKO E9.5 PEO explants (day 3 culture). Error bars indicate mean±s.e.m. (*n*=3-4). *P<0.05 (unpaired *t*-test with Welch's correction). (G,H) Immunofluorescence images (G) and corresponding quantification of the percentage Ki67+ (pro)epicardial cells (H) in control and *Srsf3* cKO E9.5 PEO explants (day 3 culture). Error bars indicate mean±s.e.m. (*n*=4 control, 3 *Srsf3* cKO embryos). *P<0.05 (unpaired *t*-test with Welch's correction). epi, epicardium; LV, left ventricle; myo, myocardium; peo, proepicardium. Scale bars: 200 µm (A); 100 µm (D,E); 50 µm (G).

the myocardium (Fig. 3B). Moreover, embryos presented a severe non-compaction phenotype at this stage (Fig. 3C). To determine whether the EMT and compaction defects may be underpinned by impaired proliferation, given the *Srsf3* cKO phenotype, we assessed epicardial cell proliferation. Epicardial explants cultured from E13.5 *Srsf3* iKO hearts revealed a mosaic loss of SRSF3, yet, as predicted, they contained fewer Ki67-positive cells (Fig. 3D,E), due to loss of proliferation where *Srsf3* was successfully deleted (Fig. 3F,G; mean 37.2% versus 1.3%, in SRSF3+ and Srsf3-depleted cells, respectively). Impaired EMT at E13.5 is consistent with the requirement for cell division to drive this process (Wu et al., 2010), further supporting the role of SRSF3 in epicardial proliferation. Moreover, defective EMT and compaction coincided with impaired sprouting of vessels from the sinus venosus in *Srsf3* iKO hearts (Fig. 3H,I), which is also in keeping with the role of the epicardium in promoting coronary vessel growth (Lupu et al., 2020a).

We then examined the major transcriptional changes that occur in *Srsf3*-depleted epicardial cells at E13.5. Whole-heart 10x Chromium single-cell RNA sequencing (scRNA-seq) was performed, rather than bulk RNA-seq, since variable recombination was expected in individual epicardial cells (Fernandez-Chacon et al., 2019; Redpath and Smart, 2021). Unsupervised graph-based clustering revealed 18 clusters, largely corresponding to discrete cardiomyocyte types, epicardial, mesenchymal, endocardial and coronary endothelial cells (Fig. 4A, Fig. S3A). The epicardial (Epi) cluster was identified based on mesothelial markers (Fig. S3A), such as *Upk3b* (Kanamori-Katayama et al., 2011; Lupu et al., 2020b). The mesenchymal (Mes) cluster was derived from the epicardium, as indicated by widespread tdTomato expression (Fig. S3B), thus essentially constituting

EPDCs. Proportionally fewer mesenchymal cells were recovered in *Srsf3* iKO hearts in comparison to controls (7.4% versus 9.6% of all cells sequenced per genotype; Fig. 4A,B), consistent with impaired EMT in *Srsf3* iKO hearts (Fig. 3B). Moreover, fluorescence-activated cell sorting (FACS)-sorted epicardial lineage cells cultured following dissociation for scRNA-seq demonstrated a mosaic loss of SRSF3 and reduced cyclin D1 expression in cells lacking *Srsf3* (Fig. 4C,D). An enrichment of the *Srsf3*-retaining proliferative population was further exaggerated over 48 h in culture, likely because *Srsf3*-depleted cells are compromised by impaired proliferation, in addition to any potential viability and adhesion defects (Fig. S3C). Despite this, we recorded an overall reduction in cyclin D1+ cells in *Srsf3* iKO and, importantly, all SRSF3-negative cells lacked cyclin D1 (Fig. 4C,D).

Through unbiased splicing analysis of *Srsf3* using scRNA-seq data, successful recombination of one or both alleles at exons 2 and 3 could be discerned in up to 10% of *Srsf3* iKO epicardial cells (Fig. S4), a figure that is likely severely under-estimated, given the 3′ bias of 10x Chromium technology, and further confounded by the failure of targeted cells to proliferate and to pass standard quality control filters. Moreover, upon assessing *Srsf3* levels in the *Srsf3* iKO epicardial population, we detected a bimodal distribution; while *Srsf3*-depleted epicardial cells were enriched within the negative to low *Srsf3*-expressing population, paradoxically a substantial proportion displayed *Srsf3* levels that exceeded those in control epicardium (Fig. 4E); this is noteworthy given the considerable decline in *Srsf3* levels that occurs ordinarily between E11.5 and E15.5 (Fig. 1A,B, Fig. S1A). Epicardial cells were thus divided into *Srsf3* negative/low or high subsets for downstream

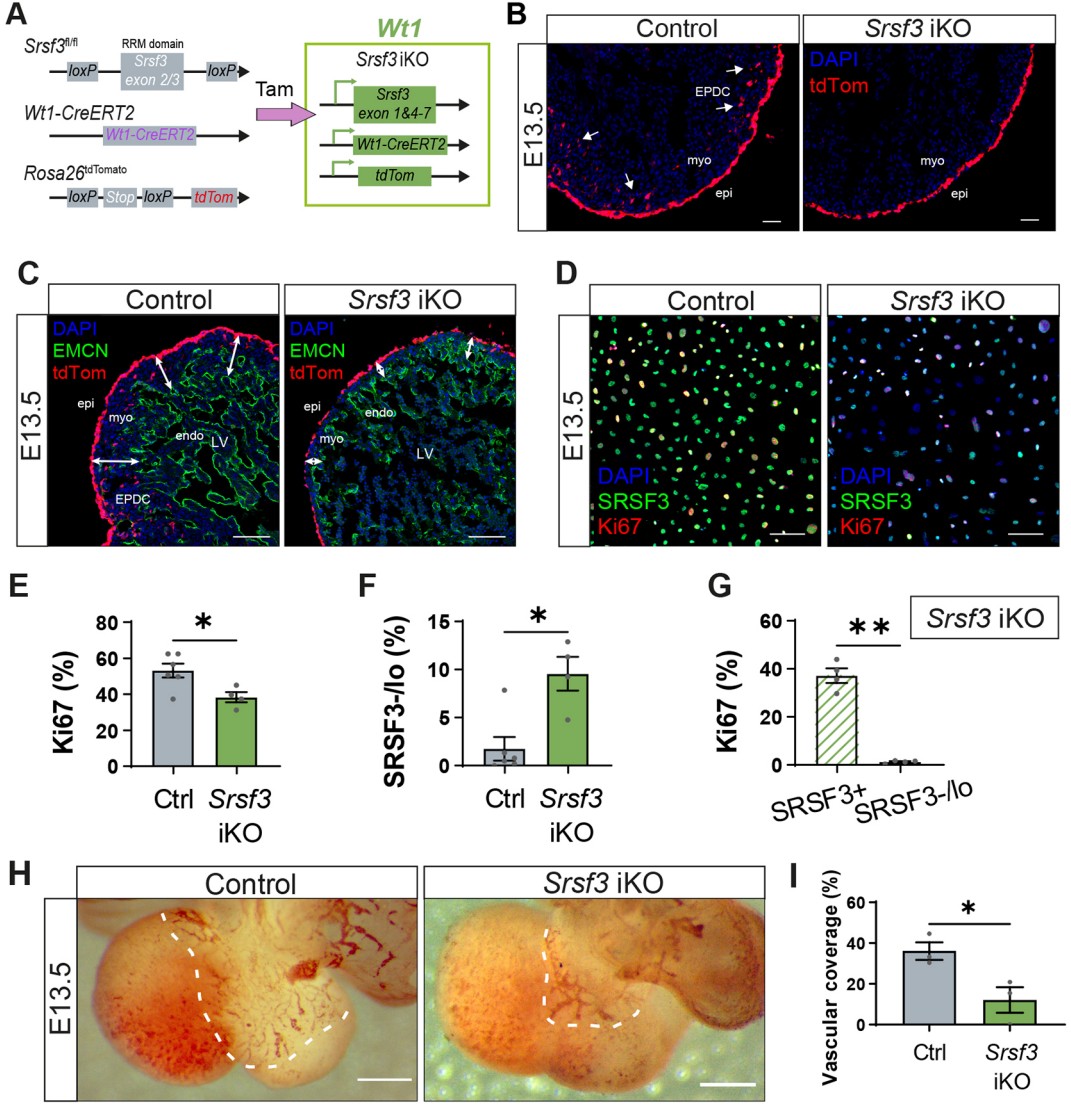

**Fig. 3. Impaired proliferation and function of epicardial cells in *Srsf3* iKO hearts.** (A) Generation of an inducible SRSF3 knockout mouse (*Srsf3* iKO) with lineage tracing capacity; restricted to the epicardial lineage (*Wt1*-CreERT2) and reported by tdTomato (Rosa26^tdTomato). Tamoxifen (Tam) was administered at E9.5 and E11.5 (late-stage induction). (B) Fluorescence images showing epicardial lineage cells (tdTomato⁺) in control and *Srsf3* iKO embryonic mouse hearts at E13.5. White arrows highlight invasion of epicardium-derived cells (EPDC) into the myocardium. Images representative of *n*=6 embryos. (C) Immunofluorescence showing the endocardium and vascular network (EMCN⁺) in control and Srsf3 iKO embryonic mouse hearts at E13.5. tdTomato expression demarcates the epicardial lineage. White arrows highlight myocardial wall thickness. Images representative of *n*=4 embryos. (D-G) Immunofluorescence images (D) and corresponding quantification of the percentage of Ki67⁺ (E) and SRSF3 negative/low (F) epicardial cells in control and Srsf3 iKO explants, and the percentage of Ki67⁺ epicardial cells as a proportion of SRSF3⁺ and SRSF3 negative/low (SRSF3−/lo) cells in *Srsf3* iKO explants (G). Error bars indicate mean±s.e.m. (*n*=4-6). *P<0.05, **P<0.01 (unpaired *t*-test with Welch's correction). (H,I) PECAM1 whole-mount DAB staining (H) and corresponding quantification of the percentage of dorsal surface area covered by sinus venosus-derived vessels (I) in control and *Srsf3* iKO embryonic mouse hearts at E13.5. Dashed lines indicate the extent of sinus venosus sprouting. Error bars indicate mean±s.e.m. (*n*=3). *P<0.05 (unpaired *t*-test with Welch's correction). endo, endocardium; EPDC, epicardium-derived cells; epi, epicardium; LV, left ventricle; myo, myocardium. Scale bars: 50 μm (B,D); 200 μm (C); 500 μm (H).

analysis. *Srsf3* negative/low epicardial cells showed an overall reduction in predicted proliferation state, with a smaller fraction of cells detected in S and G2/M phases of the cell cycle in comparison to the *Srsf3* high epicardial cells, based on their expression of cell cycle markers (Fig. 4F). Further, inferred quiescent (G0) or slow-dividing (G1) cells, distinguished by negative *Mki67* expression in the G0/G1 scored population, were enriched in the *Srsf3* negative/low epicardial population (Fig. 4G; 90.3% versus 57.9%). Importantly, our analysis implied that *Srsf3* iKO hearts had a decreased proportion of cycling epicardial cells in comparison to controls (Fig. S3C; 9.7% versus 18.8%). *Srsf3* iKO hearts also

contained more epicardial cells with upregulated expression of genes associated with quiescence, such as *Clu* (Harris et al., 2021), and senescence, for example *Map4k4*, *Tmem30a* and *Pofut2* (Kim and Kim, 2021) (Fig. 4H); accordingly, protein markers such as p21, responsible for G1/S arrest, were more abundant (Fig. 4I,J; 3.7% versus 1.2%).

Next, we performed differential gene expression analysis guided by one clustering iteration of the Epi cluster in *Srsf3* iKO hearts. This resulted in three subsets: one cluster associated with proliferation genes (Epi_G2M) and two clusters associated with epicardial genes (Epi and Epi2) (Fig. 5A). Notably, the Epi2 subset was enriched for genes

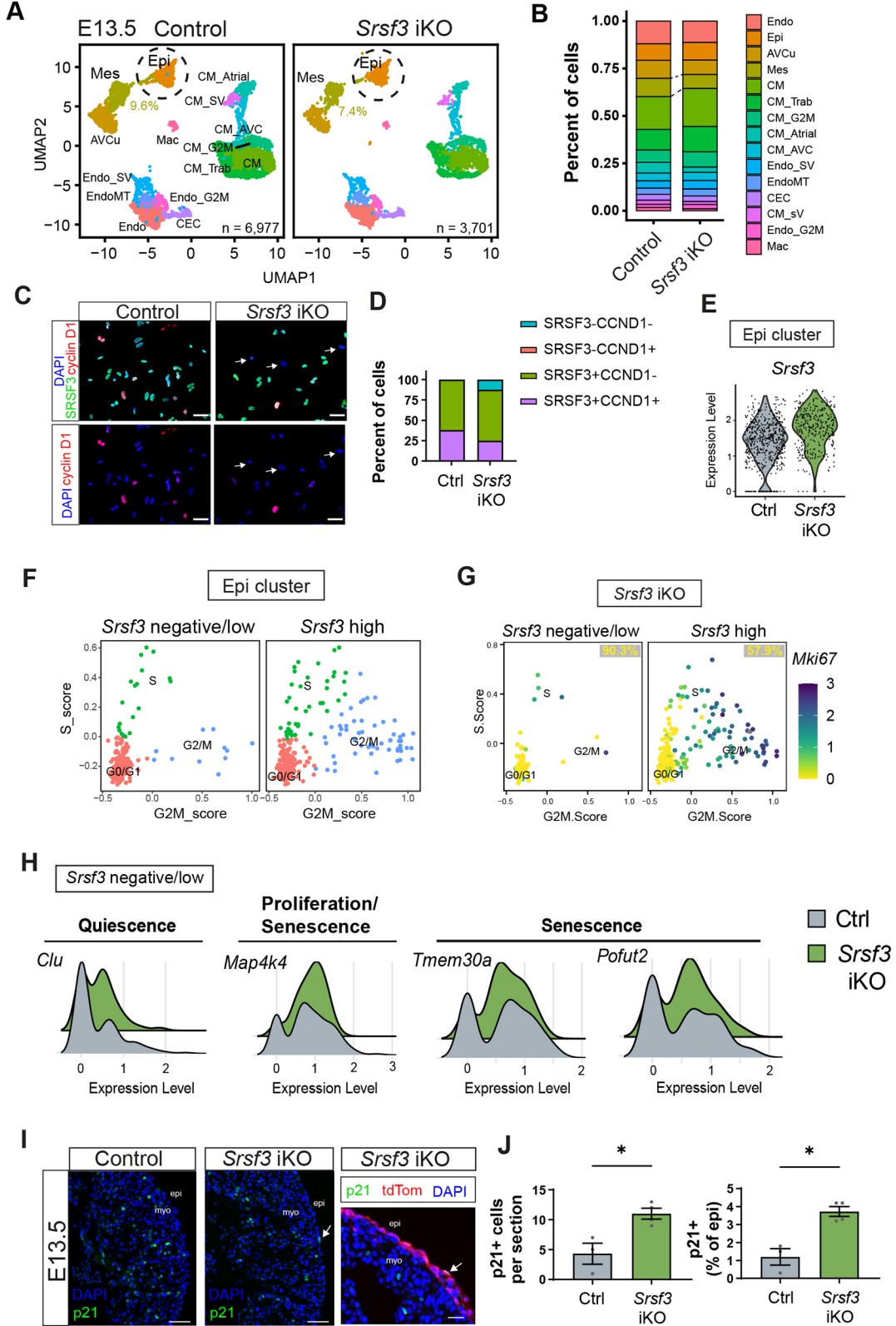

**Fig. 4.** See next page for legend.

associated with hypoxia, such as *Slc2a1* (Fig. 5A, Fig. S5A), hypoxia-induced cell death, such as *Fam162a*, and attenuated proliferation, such as *Ndrg1* (Fig. 5A,B), an expression signature that was augmented in *Srsf3* iKO epicardial cells. Indeed, the master regulator

of hypoxia, *Hif1a*, was found to be more abundant in *Srsf3* negative/low cells (Fig. 5C). Beyond the epicardial phenotype, an upregulation of hypoxia-related *Scl2a1* was detected throughout *Srsf3* iKO hearts (Fig. S5A), confirmed by elevated GLUT1 (SLC2A1) in the

**Fig. 4. Cell cycle exit and senescence of epicardial cells in *Srsf3* iKO hearts.** (A) UMAP plots showing major clusters in control and *Srsf3* iKO embryonic mouse hearts at E13.5. Chromium 10x scRNA-seq: control sample, total 6977 cells, *n*=3 hearts; *Srsf3* iKO sample, total 3701 cells, *n*=2 hearts. (B) Stacked bar plot comparing proportions of each population between control and *Srsf3* iKO hearts. (C) Immunofluorescence images showing expression of SRSF3 and CCND1 in E13.5 control and *Srsf3* iKO FACS-sorted epicardial lineage (tdTomato⁺) cells after 48 h culture. (D) Quantification of SRSF3 and CCND1 expression in FACS-sorted epicardial lineage cells from C. (E) Violin plot showing relative expression of *Srsf3* in the epicardial cluster. (F) Scatter plot showing the distribution of epicardial cells in cell cycle phases when subsetted by *Srsf3* expression. (G) Feature plot representing range of expression of *Mki67*, further separating quiescent and slow-dividing epicardial cells in *Srsf3* iKO scRNA-seq data at E13.5. Percentage of G0/G1 *Mki67* negative epicardial cells is displayed. (H) Histograms representing distribution of *Srsf3*-negative/low-expressing epicardial cells for genes related to quiescence (*Clu*), proliferation (*Map4k4*) and senescence (*Map4k4, Tmem30a* and *Pofut2*) in control and *Srsf3* iKO scRNA-seq data at E13.5. (I) Immunofluorescence imaging of p21 expression (white arrows) in control and *Srsf3* iKO hearts at E13.5; tdTomato expression demarcates the epicardium. Images representative of *n*=3 control and 4 embryos *Srsf3* iKO hearts. Arrows indicate p21-positive epicardial cells. (J) Quantification of p21⁺ epicardial cells per section, and as a percentage of total epicardial cells per heart. Error bars indicate mean ±s.e.m. (*n*=3-4). *\*P*<0.05 (unpaired *t*-test with Welch's correction). AVCu, atrioventricular cushion; CEC, coronary endothelial cells; CM, cardiomyocytes; CM_Atrial, atrial cardiomyocytes; CM_AVC, CM atrioventricular canal; CM_SV, sinus venosus cardiomyocytes; CM_Trab, trabecular cardiomyocytes; Endo, endocardial cells; EndoMT, endocardial-to-mesenchymal transition cells; Endo_SV, sinus venosus endocardial cells; epi/Epi, epicardium; _G2M, proliferating cells; Mac, macrophages; Mes, mesenchymal cells; myo, myocardium. Scale bars: 50 µm (C; I, right panel); 200 µm (I, left and middle panels).

myocardium (Fig. 5D,E, Fig. S5B), possibly due to defective coronary vasculature. To further validate the scRNA-seq data, fluorescence *in situ* hybridisation analysis showed upregulated *Ndrg1* in the epicardium of *Srsf3* iKO hearts (Fig. 5F). Increased frequency of terminal deoxynucleotidyl transferase dUTP nick end labelling (TUNEL)-positive apoptotic epicardial cells was observed in *Srsf3* iKO hearts compared to controls (Fig. 5G,H). The induction of apoptosis in *Srsf3* iKO, but not *Srsf3* cKO epicardium, may reflect the failure of SRSF3-depleted epicardial cells to cope with the hypoxic stress resulting from coronary vessel defects, causing them to undergo cell death. Given the influence of epicardial SRSF3 depletion on other parts of the heart, including hypoxic and non-compacted myocardium (Figs 3C, 5D,E), we also looked at markers previously associated with ventricular non-compaction (Rhee et al., 2021). Stress-associated natriuretic peptide genes, *Nppa* and *Nppb*, were upregulated in all cardiomyocyte subsets, concomitant with downregulation of *Ttn* (contractility) and *Fabp3* (metabolism), and a reduction in the proliferation markers *Top2a* and *Pcna* selectively in cycling cardiomyocytes (Fig. S6A-C). Collectively, these data suggest that SRSF3 regulates epicardial proliferation and survival, to uphold epicardial function through EPDC contribution, promotion of coronary vessel growth and ventricular wall compaction.

## SRSF3 regulates cell cycle progression in epicardial cells

To investigate further the role of SRSF3 in epicardial cell proliferation and circumvent problems associated with *in vivo* administration of tamoxifen, the active form of tamoxifen, 4-hydroxytamoxifen (4-OHT), was used to induce gene deletion *in vitro* in E11.5 epicardial explants (Fig. 6A). Direct treatment resulted in fewer epicardial cells (Fig. 6B) and smaller outgrowth (Fig. 6C) from *Srsf3* iKO explants compared to Wt1^CreERT2^;Rosa26^TdTomato^; *Srsf3*^+/+^ controls. While this strategy enabled more controlled targeting of cells, staining for SRSF3

still revealed mosaic depletion (Fig. 6D). In successfully targeted cells, (SRSF3-/lo), Ki67 expression was diminished (Fig. 6D,E), reinforcing the finding that the proliferative capacity of SRSF3-depleted epicardial cells is impaired.

## SRSF3 coordinates epicardial proliferation by controlling key cell cycle regulators and upstream stimulatory pathways

To understand the mechanisms through which SRSF3 controls epicardial proliferation, we undertook an unbiased analysis of SRSF3-dependent RNA processing, by performing bulk RNA-seq in control or *Srsf3* knockdown MEC1 epicardial cells, a murine embryonic line derived by Li et al. (2011) (Fig. 7A). We found that 2680 and 2856 genes were significantly up- and downregulated, respectively, in *Srsf3*-depleted cells (Fig. 7B, Table S2). Gene Ontology (GO) enrichment analysis identified 'metabolic and signal transduction processes' as the most significantly affected biological function, followed by 'regulation of cell cycle phase transition', and terms related to mitosis and nuclear division (Fig. S7, Tables S3-S12). Guided by the known role of Srsf3 as a splicing factor, we next conducted a detailed analysis of differential RNA splicing, which revealed 10,199 splicing events significantly altered upon *Srsf3* depletion. The majority of *Srsf3*-dependent alternative splicing events involved skipped exons (64.5%; Table S13, Fig. 7C,D), with fewer alterations at the level of differentially retained introns (7.1%; Table S14), mutually exclusive exons (14.6%; Table S15), or alternative 5′ (5.6%; Table S16) or 3′ splice sites (8.2%; Table S17). Of the *Srsf3*-dependent skipped exon events, more +ΔPsi transcripts were detected [exon skipped more often in *Srsf3* knockdown samples; 4038 (61.4%)] than −ΔPsi [exon included more often in *Srsf3* knockdown samples; 2539 (38.6%); Fig. 7C]. GO term enrichment analysis of alternatively spliced transcripts shows that *Srsf3* knockdown affects many cellular functions, including splicing itself, as well as multiple genes that regulate the cell cycle (Tables S8-S12). We additionally detected 338 significant incidences of differential polyadenylation site usage (PAU) events following *Srsf3* depletion, although the reproducibility between independent siRNAs was limited when filtered at *P*<0.05 and minimum 20% PAU difference (25 events considered significant only with #1, 283 with #2, 30 significant with both oligos; Table S18). Taken together, SRSF3 impacts epicardial cell cycle via regulation of transcription and alternative splicing, and may additionally have a role in polyadenylation of epicardial genes.

To further substantiate the role of SRSF3 in epicardial RNA processing, we comprehensively mapped the SRSF3-RNA interaction networks across the epicardial transcriptome, by infrared CLIP (irCLIP), an ultra-efficient variation of the UV-C crosslinking immunoprecipitation (CLIP) protocol combined with high-throughput sequencing (Zarnegar et al., 2016), using a mouse immortalised epicardial cell line (Austin et al., 2008) (Fig. 8A, Fig. S8A,B). Significant crosslink sites (FDR<0.05, with short peaks <5 nt wide excluded) defined 47,273 peaks of SRSF3 binding mapped to 8617 different genes across the epicardial transcriptome, with a consensus binding motif matching the one identified in previous SRSF3 iCLIP studies (Fig. 8B) (Änkö et al., 2012; Müller-McNicoll et al., 2016; Ratnadiwakara et al., 2018). SRSF3 was found to bind non-coding as well as protein-coding transcripts, with binding sites distributed across predominantly coding sequences and introns, but also regulatory regions (Fig. 8C). Consistent with SRSF3 iCLIP in murine P19 cells (Änkö et al., 2012), exonic binding was enriched within 100-200 nt of 5′ and 3′ splice sites (Fig. S8C).

GO analysis of genes with significant SRSF3 binding peaks demonstrated an enrichment in functions relating to mitotic cell

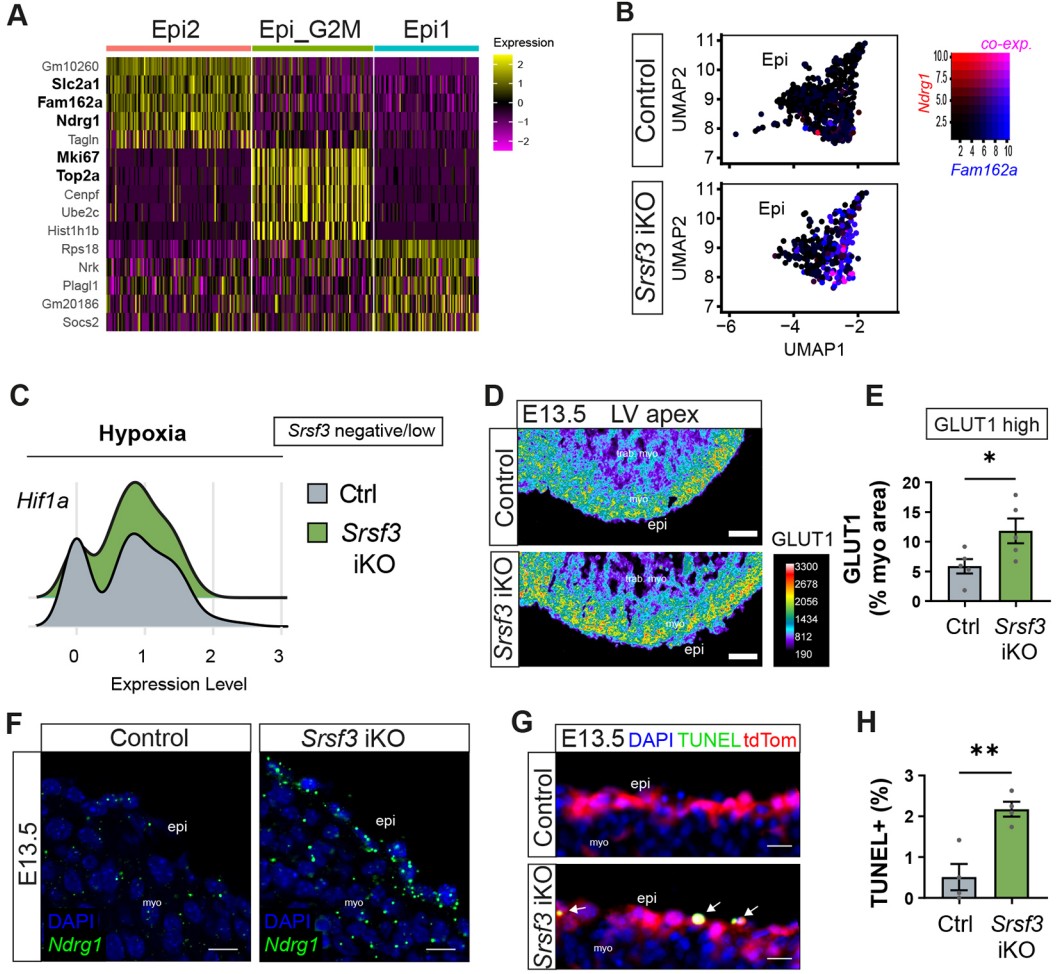

**Fig. 5. Upregulated hypoxia and increased cell death of epicardial cells in *Srsf3* iKO hearts.** (A) Heatmap of the top 5 differentially expressed genes for each cluster following one clustering iteration of the Epi cluster in *Srsf3* iKO hearts. High expression is indicated in yellow. Genes of interest are highlighted in bold. (B) Feature plots representing range of (co)expression of *Ndrg1* and *Fam162a* mRNA in individual cells of the epicardial (Epi) cluster. (C) Histograms representing distribution of *Srsf3*-negative/low-expressing epicardial cells for *Hif1a* in control and *Srsf3* iKO scRNA-seq data at E13.5. (D,E) Pseudocoloured immunofluorescence images (D) and corresponding quantification of GLUT1 high expression as percentage of the ventricular myocardial area (E) in control and *Srsf3* iKO embryonic mouse hearts at E13.5. Error bars indicate mean±s.e.m. (*n*=5). **P*<0.05 (unpaired *t*-test with Welch's correction). (F) Fluorescence *in situ* hybridisation of *Ndrg1* mRNA in the epicardium of control and *Srsf3* iKO mouse hearts at E13.5. Images representative of *n*=4 embryos. (G,H) TUNEL assay (G) and corresponding quantification of percentage apoptotic (TUNEL+) epicardial cells (H) in control and *Srsf3* iKO embryonic mouse hearts at E13.5. White arrows highlight apoptotic epicardial cells. Error bars indicate mean±s.e.m. (*n*=4). ***P*<0.01 (unpaired *t*-test with Welch's correction). epi, epicardium; LV, left ventricle; myo, myocardium; trab, trabecular. Scale bars: 75 μm (D); 10 μm (F); 20 μm (G).

cycle, RNA metabolism and intracellular transport (Fig. 8D), in keeping with known functions of SRSF3 (Änkö et al., 2012; Müller-McNicoll et al., 2016; Ratnadiwakara et al., 2018). Among the prominent cell cycle regulators bound by SRSF3 were *Mki67*, *Cdk1* and *Ccnd1* (Table S19). Focussing on SRSF3-bound transcripts classified as 'positive regulators of G1/S and G2/M transition of mitotic cell cycle' (Tables S20 and S21) highlighted canonical regulators, including cyclins, cell division control (Cdc) genes and cyclin-dependent kinases, such as *Ccnd1*, *Cdc7*, *Cdc25a-c*, *Cdk4* and *Cdk1*, as well as upstream inducers of proliferation, such as *Egfr*, *Akt1* and *Camk2d* (Fig. 8E). Protein interaction (STRING) analysis of these gene products indicated a functional association network of SRSF3-regulated targets, centred around cyclin D1 (Fig. 8F). The murine *Ccnd1* gene, comprising five exons, yields a single protein-coding mRNA (*Ccnd1-201*) and the prominent peaks of SRSF3 binding mapped to exons 1, 2, 3 and 4 (Fig. 8G). While splice variants of human *CCND1* are associated with

hyperproliferation and malignancy (Comstock et al., 2009), there is no evidence for expression of alternatively spliced protein-coding *Ccnd1* transcripts in mice (Augello et al., 2015; Rojas et al., 2009). Therefore, the exonic SRSF3 binding identified by irCLIP may rather relate to non-splicing roles in RNA processing, to influence transcription or stability. Indeed, *Ccnd1* transcript (Figs 5B, 8H, Fig. S8D,E) and cyclin D1 protein levels (Fig. 8I) were significantly reduced in *Srsf3* knockdown epicardial cells, compared with controls, supporting the finding that SRSF3 depletion induces cell cycle arrest. To distinguish between a primary role in transcriptional control or stability, we conducted mRNA stability assays, as previously described (Ratnadiwakara and Änkö, 2018). After inhibiting transcription with actinomycin D and measuring *Ccnd1* decay, alongside that of the G1/S regulator *Ccng1*, we found no significant differences in half-life or decay rate constant for either gene after knockdown of *Srsf3* (Fig. S9). Collectively, these data support a direct role for SRSF3 in controlling cyclin D1

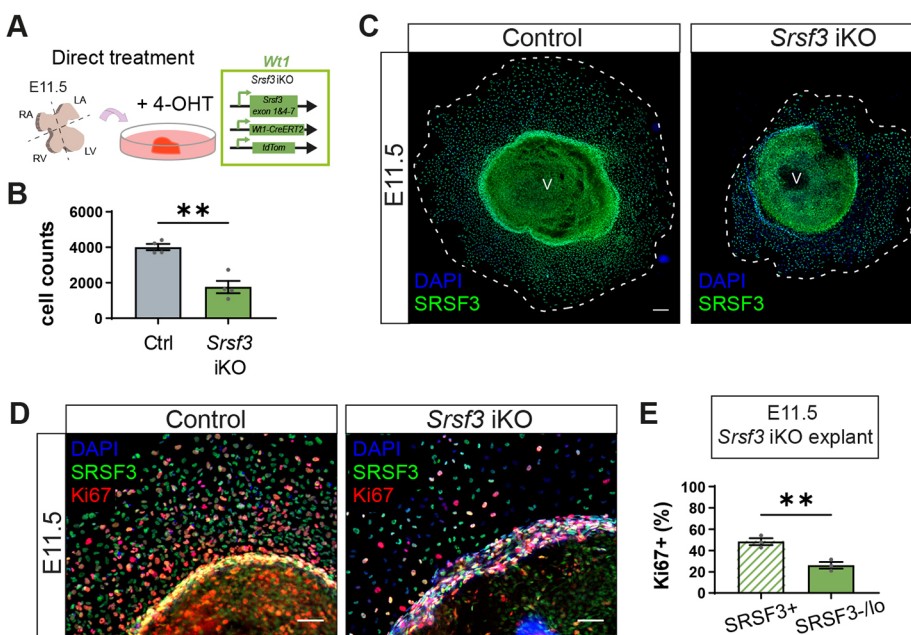

**Fig. 6. SRSF3 regulates cell cycle progression in primary epicardial cell cultures.** (A) Alternative strategy to induce SRSF3 deletion in E11.5 epicardial explants. (B) Quantification of epicardial cells from control and *Srsf3* iKO explants (day 3 culture). Error bars indicate mean ±s.e.m. (*n*=4). **P<0.01 (unpaired *t*-test with Welch's correction). (C) Immunofluorescence images showing expression of SRSF3 in control and *Srsf3* iKO explants. Dashed lines delineate the extent of epicardial outgrowth. Image representative of *n*=4 embryos. (D,E) Immunofluorescence images (D) and corresponding quantification of the percentage Ki67+ epicardial cells as a proportion of SRSF3+ and SRSF3 negative/low (SRSF3−/lo) cells (E) in *Srsf3* iKO explants. Error bars indicate mean±s.e.m. (*n*=3). **P<0.01 (unpaired *t*-test with Welch's correction). LA, left atrium; LV, left ventricle; RA, right atrium; RV, right ventricle; V, ventricle. Scale bars: 200 µm (C); 100 µm (D).

transcription, over and above any additional regulation via the upstream pathways identified by irCLIP (Table S19), defining SRSF3 as a key orchestrator of epicardial proliferation.

To further understand the mechanisms controlling epicardial cell cycle exit, we homed in on SRSF3 targets implicated in cellular senescence (Figs 4H, 9A). Specifically, a subset of genes identified by irCLIP are associated with senescence-associated secretory phenotype (SASP), oxidative stress induced senescence, DNA damage/telomere stress induced senescence, formation of senescence-associated heterochromatin foci (SAHF) and oncogene-induced senescence (Fig. 9B, Tables S22 and S23), via networks in which c-Jun and p53 are central regulators, downstream of MAP kinase signalling (Fig. S10) (Deng et al., 2023). While the diversity of targets infers multiple levels of control over cellular senescence by SRSF3, we further investigated *Map4k4*, a 31-exon gene which, in cancer cells, is alternatively spliced by SRSF3 to control c-Jun activation and regulate proliferation (Lin et al., 2018). In keeping with this, we detected significant alternative splicing of *Map4k4* after knockdown of *Srsf3* (Fig. S11A, Tables S13, S15, S17). Eighteen alternatively spliced transcripts of *Map4k4* are reported to exist in human and mouse, via the selective skipping or inclusion of exons 16, 17, 21 and 24, with at least 11 believed to encode proteins (Ensembl). irCLIP revealed SRSF3 binding across the length of the gene (Fig. S11B), with notable peaks close to intron-exon boundaries (Fig. 9C). Although total levels of *Map4k4* were unaffected (Fig. 9D), *Srsf3* depletion resulted in significant alternative splicing, in favour of an overall increase in the skipping of exon 17, a key exon implicated from cancer studies in the control of proliferation (Fig. 9E,F). Specifically, in the absence of *Srsf3*, exon 16 skipped:exon 17 retained isoforms (*Map4k4-206*, *-208*, *-209*) were reduced, whilst exon 16 retained:exon 17 skipped isoforms (*Map4k4-202*, *-215*, *-217*) were increased, as were isoforms lacking both exons 16 and 17 (*Map4k4-205*, *-213*). Isoforms with both exons retained (*Map4k4-201*, *-218*) were lowly expressed in MEC1 cells at baseline, and unchanged by *Srsf3* knockdown. Taken together, these data demonstrate a role for SRSF3 in the alternative splicing of *Map4k4*, to alter the balance between isoforms that are known to preferentially promote proliferation (Jovanovic et al., 2022) or senescence (Zheng et al., 2024), suggesting that regulation by

SRSF3 may, in part, contribute to the switch of epicardial cells from the proliferative to senescent state.

## Non-recombined epicardial lineage cells compensate for loss of SRSF3-depleted cells

Having identified defects in epicardial proliferation and viability in *Srsf3*-depleted cells, contrasting with the retention or upregulation of *Srsf3* levels in non-recombined cells at E13.5, we sought to determine the impact of mosaic targeting on later stage heart development. We first assessed the tdTomato-positive epicardial lineage of *Srsf3* iKO and control hearts at E15.5 using SMART-Seq2 (Picelli et al., 2014). Surprisingly, by this stage, the *Srsf3* iKO lineage displayed similar composition to controls, consisting of a small epicardial cluster, two mesenchymal clusters and a distinct cluster of mural cells (Fig. 10A, Fig. S12A). We previously characterised the Mes1 cluster as subepicardial mesenchyme and Mes2 as fibroblast-like cells (Lupu et al., 2020b). Since SMART-Seq2 allows recovery of full-length mRNA, we could establish that no recombined cells lacking *Srsf3* exons 2/3 were present at this stage, indicating that *Srsf3*-depleted cells must have been eliminated earlier in development. As before, tdTomato expression did not accurately correlate with gene depletion, due to differing recombination efficiencies. Therefore, to confirm that only non-*Srsf3*-recombined cells remained, E15.5 explants were stained for SRSF3 (Fig. S12B). In contrast to E13.5 explants, all outgrowing epicardial cells at E15.5 had high levels of SRSF3, confirming that the SRSF3-depleted cells had been removed earlier in development. In fact, the SMART-Seq2 data paradoxically indicated increased *Srsf3* expression in both epicardial and subepicardial mesenchymal cells of *Srsf3* iKO hearts, compared with controls, which was confirmed by SRSF3 IHC (Fig. 10B,C, Fig. S12C). Thus, the modest elevation of *Srsf3* in non-recombined iKO epicardial cells initially detected at E13.5 is robustly maintained, and even enhanced, relative to control cells, across a developmental window through to at least E15.5. SRSF3 upregulation in non-recombined E15.5 iKO epicardial cells coincided with increased proliferative capacity, as indicated by *Top2a* expression and confirmed by TOP2α staining in the epicardium (Fig. 10D,E,

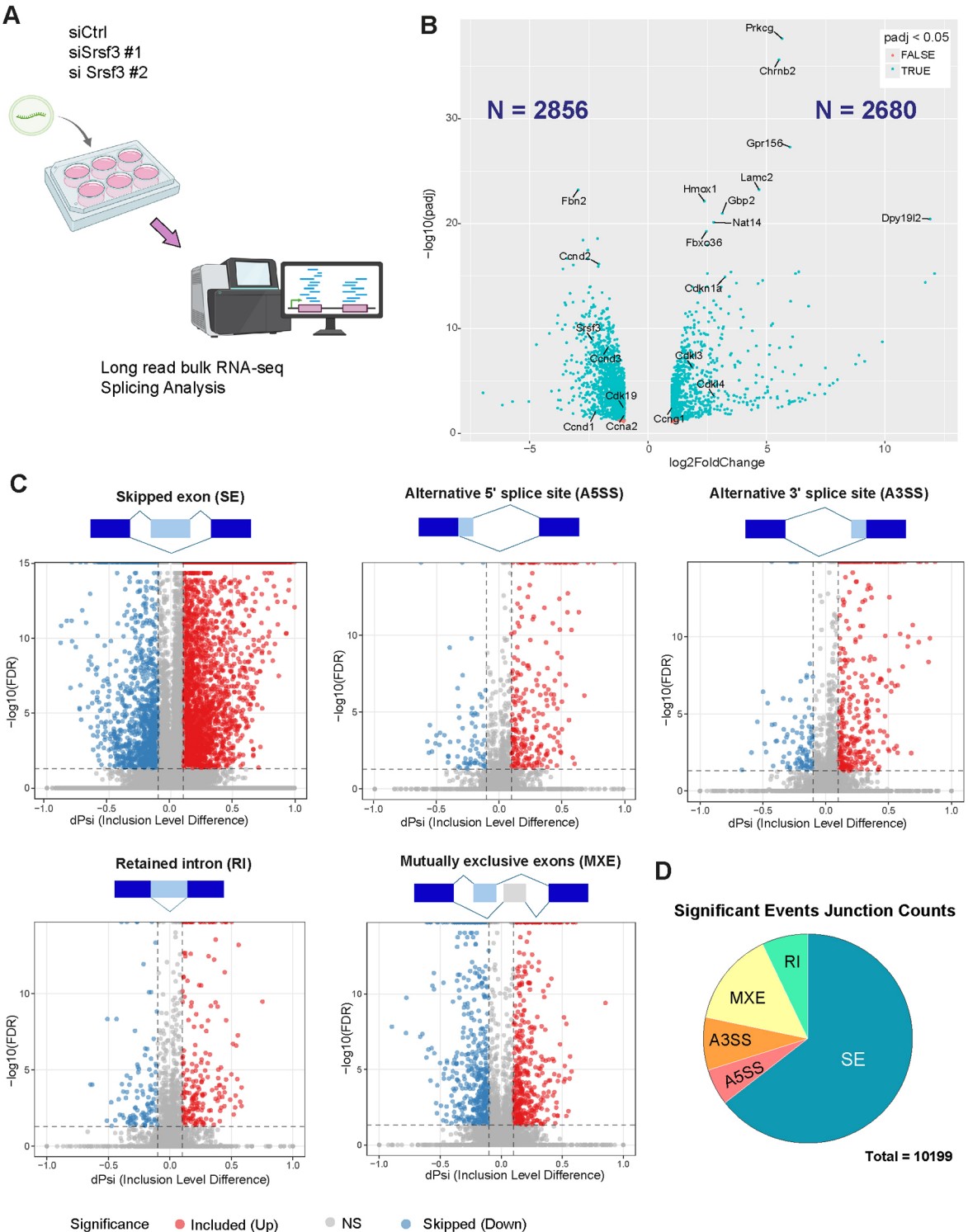

**Fig. 7. SRSF3 controls the alternative splicing of epicardial cell cycle regulatory genes.** (A) Schematic of transfection and RNA-seq workflow for transcriptomic analysis of *Srsf3* knockdown in MEC1 epicardial cell line. (B) Volcano plot depicting differentially expressed genes, with prominent cell cycle regulators and the top 10 altered genes with highest adjusted *P*-value annotated. (C) Volcano plots to show all significant *Srsf3*-dependent alternative splicing events, including 6577 skipped exons, 575 alternative 5′ splice sites, 832 alternative 3′ splice sites, 725 retained introns and1490 mutually exclusive exons. Positive dPsi values indicate that the exon is preferentially skipped in knockdown (inclusion is higher in control). (D) Pie chart summarising the proportions of significant alternative splicing events: 64.5% skipped exons (SE); 5.6% alternative 5′ splice sites (A5SS); 8.2% alternative 3′ splice sites (A3SS); 7.1% retained introns (RI) and 14.6% mutually exclusive exons (MXE).

Fig. S12D). Taken together, these data imply that the *Srsf3*-retaining epicardial cells that evaded recombination are more proliferative and rapidly outnumber *Srsf3*-depleted cells undergoing quiescence or senescence. This appears not to occur via the archetypal MYC-mediated cell competition mechanism (Villa Del Campo et al., 2016), given the scarcity of *Myc* in the epicardium (Fig. S12E,F)

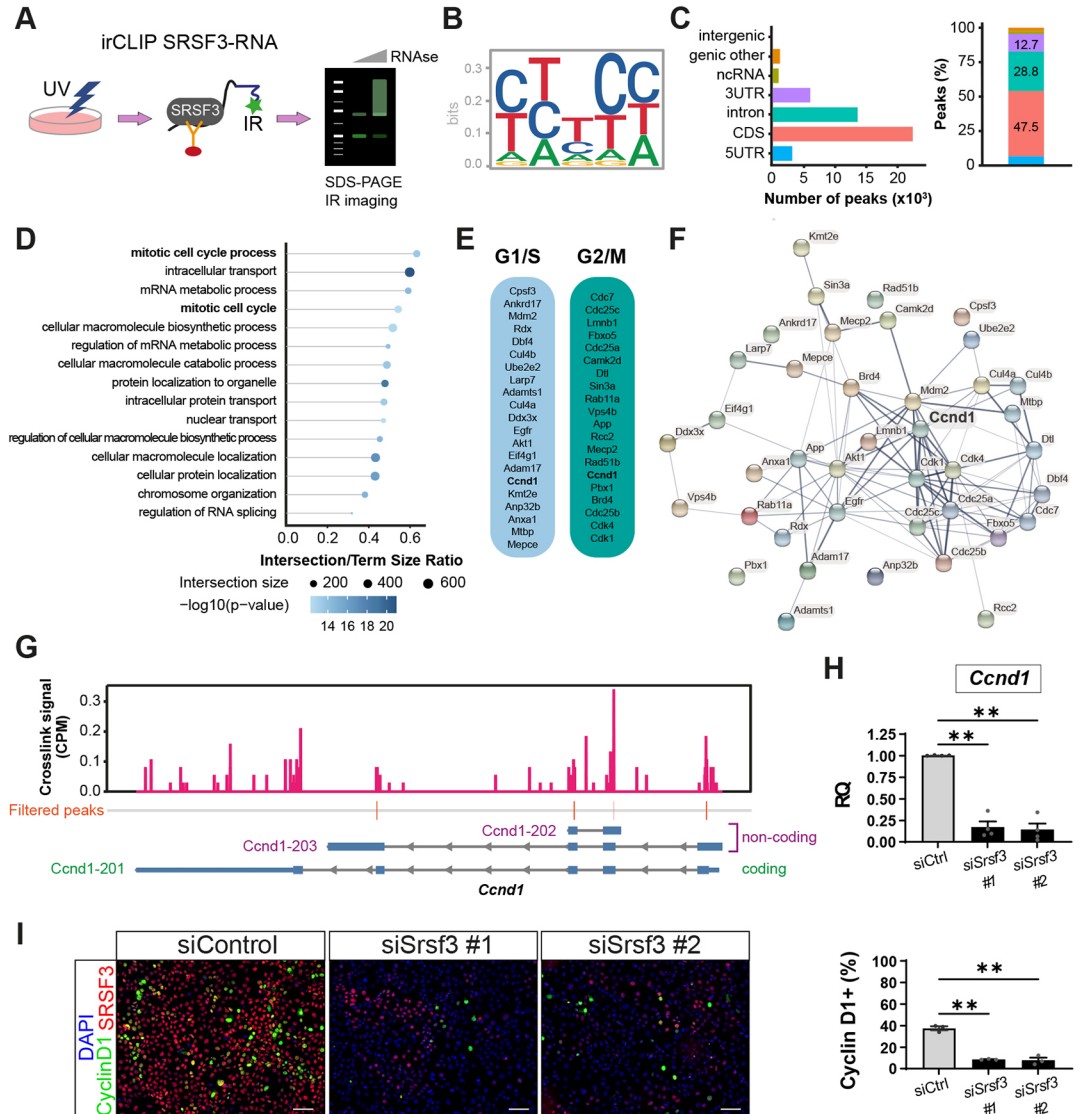

**Fig. 8. SRSF3 orchestrates proliferation by binding multiple cell cycle regulatory genes and controlling *Ccnd1* transcript levels in epicardial cells.** (A) Schematic of irCLIP workflow to map direct trancriptome-wide SRSF3–RNA interactions in the epicardial cell line. IR, infrared; UV, ultraviolet. (B) SRSF3 consensus binding motif in epicardial cells. (C) Number and percentage of significant SRSF3 crosslink peaks (FDR<0.05) across the transcriptome. (D) GO term analysis on RNA targets identified showing the most over-represented biological processes. (E,F) SRSF3 binding detected to RNAs transcribing 'positive regulators of G1/S and G2/M transition of mitotic cell cycle' (E) and corresponding functional association network of these SRSF3 RNA targets in epicardial cells (F). *Ccnd1*, common to both G1/S and G2/M transition, occupies a central node within the pro-mitotic regulatory network. (G) SRSF3 irCLIP binding profile within *Ccnd1* transcript. Significant peaks, most likely to represent 'functional' binding sites were filtered to remove those <5 nt wide (likely transient binding). *Ccnd1* variants are shown (Ccnd1-201, protein coding; Ccnd1-202 and Ccnd1-203, retained intron). (H) qRT-PCR analysis of total *Ccnd1* transcripts in control and *Srsf3* siRNA-transfected cell lines. RQ, relative quantification. Error bars indicate mean±s.e.m. (*n*=4). (I) Immunofluorescence images and corresponding quantification of percentage CCND1+ epicardial cells in control and *Srsf3* siRNA-transfected cell lines. Scale bars: 100 μm. Error bars indicate mean±s.e.m. (*n*=3). **P<0.01 (one-way Welch ANOVA and Dunnett's multiple comparison test).

and its unaltered expression upon *Srsf3* knockdown (Table S2), even though SRSF3 has previously been shown to control MYC expression in cancer (Xiong et al., 2022). Further, we could detect no evidence for the Hippo signalling pathway controlling epicardial cell number via contact inhibition, with transcription of *Yap1*, *Taz* (*Wwtr1*) or *Tead1-4* (Fig. S12G) and active (non-LATS1/2 phosphorylated), nuclear-localised YAP1 protein (Fig. S12H,I) unaltered.

Coinciding with hyperproliferation of *Srsf3*-retaining iKO epicardial cells, we found that the reduced incidence of EPDCs invading the myocardium from E13.5 through to E15.5 (Figs 3B, 10F, Fig. S12J) was fully restored by E17.5 (Fig. 10G, Fig. S12J),

and the earlier compaction defect rescued, to leave hearts appearing morphologically normal by this late stage in development (Fig. S12K). Together, these data indicate that the epicardial cells that evade recombination possess a remarkable capacity to compensate for the early loss of SRSF3-depleted cells, by using SRSF3-controlled mechanisms to promote epicardial proliferation and restore normal function.

To investigate the molecular basis for *Srsf3* upregulation in non-recombined *Srsf3* iKO epicardial cells, we investigated its splicing in MEC1 epicardial cells, since SR proteins tightly auto-regulate their expression via selective inclusion of a 'poison cassette exon' that promotes transcript degradation by nonsense-mediated decay

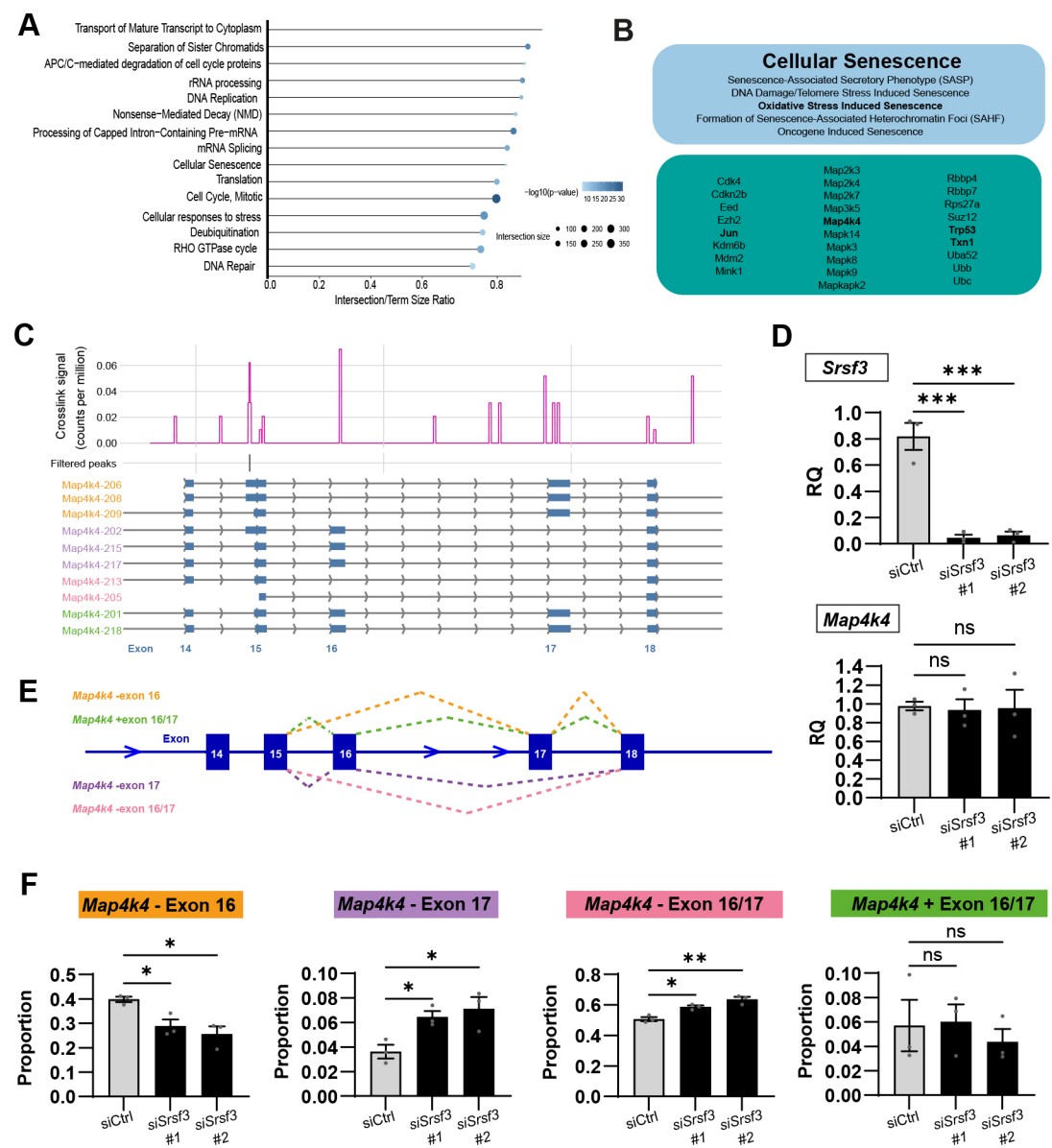

**Fig. 9. SRSF3 controls the alternative splicing of *Map4k4*, a gene that regulates cell proliferation and senescence.** (A) Reactome analysis on RNA targets identified by irCLIP, showing highly represented biological processes. (B) SRSF3 target genes associated with cellular senescence (sub)terms. Genes and processes of interest are highlighted in bold. (C) SRSF3 irCLIP binding profile across exons 14-18 of the *Map4k4* gene, to highlight SRSF3 binding sites around key exons 16 and 17, and the protein coding isoforms subject to alternative splicing in this region. (D) qRT-PCR analysis of *Srsf3* and total *Map4k4* transcripts in control and *Srsf3* siRNA-transfected MEC1 cell line. RQ, relative quantification. (E) Schematic of splicing events assessed by qRT-PCR after *Srsf3* knockdown. (F) qRT-PCR analysis of *Map4k4* splicing in control and *Srsf3* siRNA-transfected MEC1 cell line. Error bars indicate mean ±s.e.m. (*n*=3 separate experiments). *$P<0.05$, **$P<0.01$ (one-way Welch ANOVA and Dunnett's multiple comparison test). ns, not significant.

(Änkö et al., 2010; Jumaa and Nielsen, 1997; Ni et al., 2007). While control-transfected MEC1 included auto-regulatory exon 4 in 10.66% of *Srsf3* transcripts, exon 4 was present in 1.67% (siSrsf3#1) and 3.09% (siSrsf3#2) of transcripts from *Srsf3* knockdown MEC1 cells (Fig. S13A). Consistent with the notion of auto-regulation, we detected extensive SRSF3 CLIP-tag clusters across the autoregulatory cassette exon 4 of *Srsf3* (Fig. S13B), supporting a mechanism for *Srsf3* auto-upregulation in surviving cells following *Srsf3* iKO or knockdown. In contrast, the autoregulatory exon of *Srsf4*, which is known not to be cross-regulated by SRSF3 (Änkö et al., 2012), was not differentially spliced in *Srsf3* knockdown MEC1 cells (Fig. S13A). Beyond SRSF3, SRSF1 and polypyrimidine tract-binding protein (PTBP) 1

and 2 can also regulate *Srsf3* expression by promoting exon 4 skipping, yet this is not indicated, at least at the level of their expression in *Srsf3* iKO epicardial cells (Fig. S13C), collectively supporting auto-regulation by SRSF3 itself as the predominant mechanism driving elevated expression and hyperproliferation.

## SRSF3 depletion in epicardial progenitors leads to compaction and coronary vasculature defects

To explore the extent to which compensatory mechanisms could overcome more severe early epicardial deficiencies, we tested two alternative strategies to achieve more efficient gene deletion in the *Srsf3* iKO model with a higher dose of tamoxifen (80 mg/kg) at (1) E8.5, expected to target progenitors in

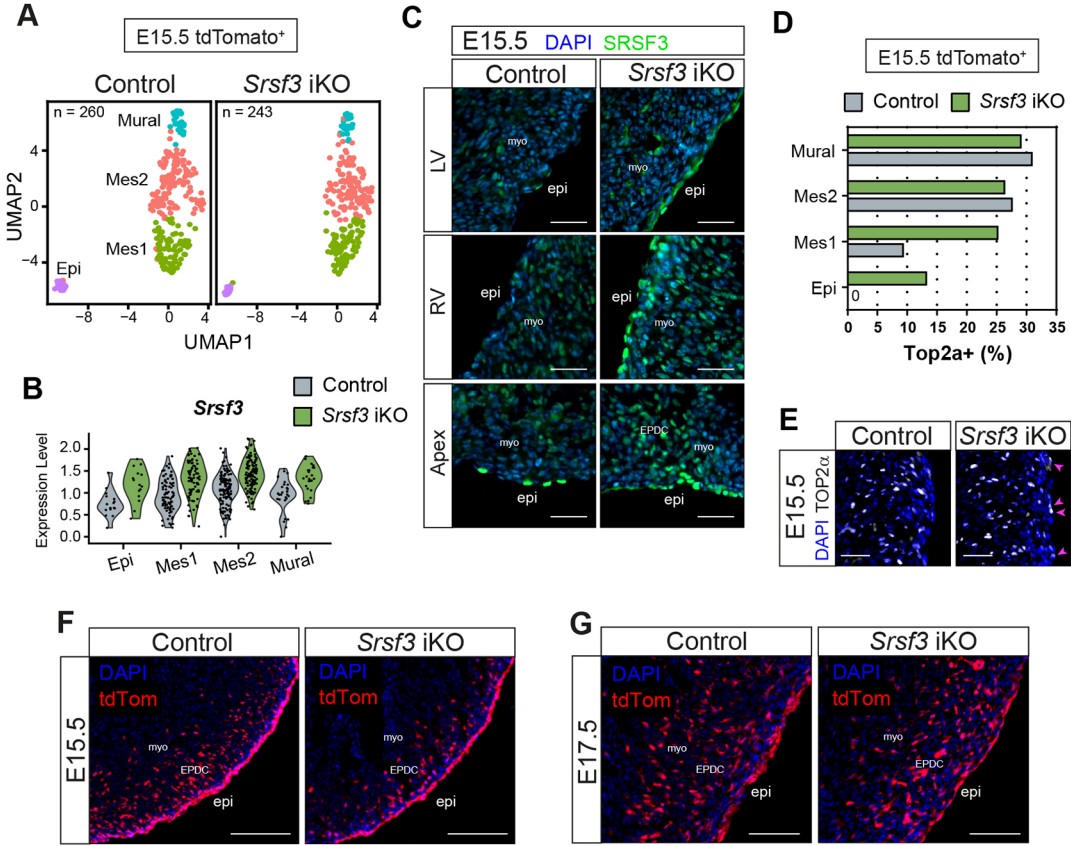

**Fig. 10. Non-recombined cells of the epicardial lineage upregulate SRSF3 and hyperproliferate to compensate for loss of SRSF3-depleted cells in *Srsf3* iKO hearts.** (A) UMAP plots showing the major cell clusters in the epicardial lineage of control and *Srsf3* iKO embryonic mouse hearts at E15.5. FACS-sorted epicardial lineage (tdTomato⁺) cells plate-based scRNA-seq: control sample, total 260 cells, *n*=6 hearts; *Srsf3* iKO sample, total 243 cells, *n*=4 hearts. (B) Violin plot showing relative expression of *Srsf3* in individual clusters. (C) Immunofluorescence images showing expression of SRSF3 in the epicardium of control and *Srsf3* iKO embryonic mouse hearts at E15.5. Images representative of *n*=4. (D) Quantification of the percentage *Top2a*-expressing epicardial lineage cells in individual clusters of control and *Srsf3* iKO scRNA-seq data at E15.5. (E) Immunofluorescence images showing expression of TOP2α in the epicardium of control and *Srsf3* iKO embryonic mouse hearts at E15.5. Magenta arrowheads highlight upregulated expression of TOP2α in epicardial cells. Images representative of *n*=3. (F,G) Fluorescence images showing epicardial lineage cells (tdTomato⁺) in control and *Srsf3* iKO embryonic mouse hearts at E15.5 (F) and E17.5 (G). Images representative of *n*=3 embryos. epi, epicardium; EPDC, epicardium-derived cells; Epi, epicardial cells; Mes, mesenchymal cells; myo, myocardium. Scale bars: 50 μm (C,E); 100 μm (F,G).

the PEO; and (2) E9.5, expected to improve targeting of epicardial founders. We observed that induction at E8.5 resulted in a more severe myocardial non-compaction in some hearts, as revealed by the persistence of a highly trabeculated EMCN⁺ endocardium, which extended closer to the epicardial surface (Fig. 11A, Fig. S14A). Given the variable extent of compensation and limited accuracy of histological morphometric analysis, this phenotype was not statistically significant overall (Fig. 11B). However, the vascular defects remained at E17.5 after E8.5 induction, evidenced by an increased proportion of superficial vessels, relative to those that had invaded the myocardium, in *Srsf3* iKO hearts compared to controls (quantified as percentage of lumenised vessels within the 50 μm of myocardium underlying the epicardium; Fig. 11A,C, Fig. S14, magenta arrows). Impaired coronary vasculature formation, notably the appearance of ectopic subepicardial vessels, frequently accompanies myocardial non-compaction in hearts with compromised epicardial function (Phillips et al., 2008; Smart et al., 2007; Tian et al., 2013). Although epicardial formation and function was initially impaired with all tamoxifen regimes tested, each showing disrupted morphology, with rounded cells on the surface and limited invasion, the respective E17.5 phenotypes appear to reflect the

differential capacity for compensation, which was greater in the E9.5/E11.5 regime, compared with E8.5. Efficient compensation and phenotypic rescue was similarly observed in *Srsf3* iKO hearts with 80 mg/kg tamoxifen at E9.5, comparable to the E9.5/E11.5 phenotype (Fig. S14), suggesting that timing of administration determines the extent of *Srsf3* recombination and functional impairment. Taken together, these data indicate that the transient functional impairments observed with the original tamoxifen regime persist when a higher efficiency SRSF3 depletion is achieved (Fig. 11, Fig. S14), due to the essential role of cell division in epicardial function (Wu et al., 2010).

## DISCUSSION
Our study presents insight into the crucial role of SRSF3 in mediating proliferation of the (pro)epicardium, to enable proper heart development. We show that SRSF3 expression is widespread in embryonic hearts, with highest levels detected in proepicardial cells and in the epicardial layer until E12.5. Constitutive SRSF3 depletion in epicardial progenitor cells resulted in gross morphological heart abnormalities and embryonic lethality at E12.5. Failure to form the epicardial layer was due to a diminished source of epicardial progenitor cells within the PEO, as a result of their decreased

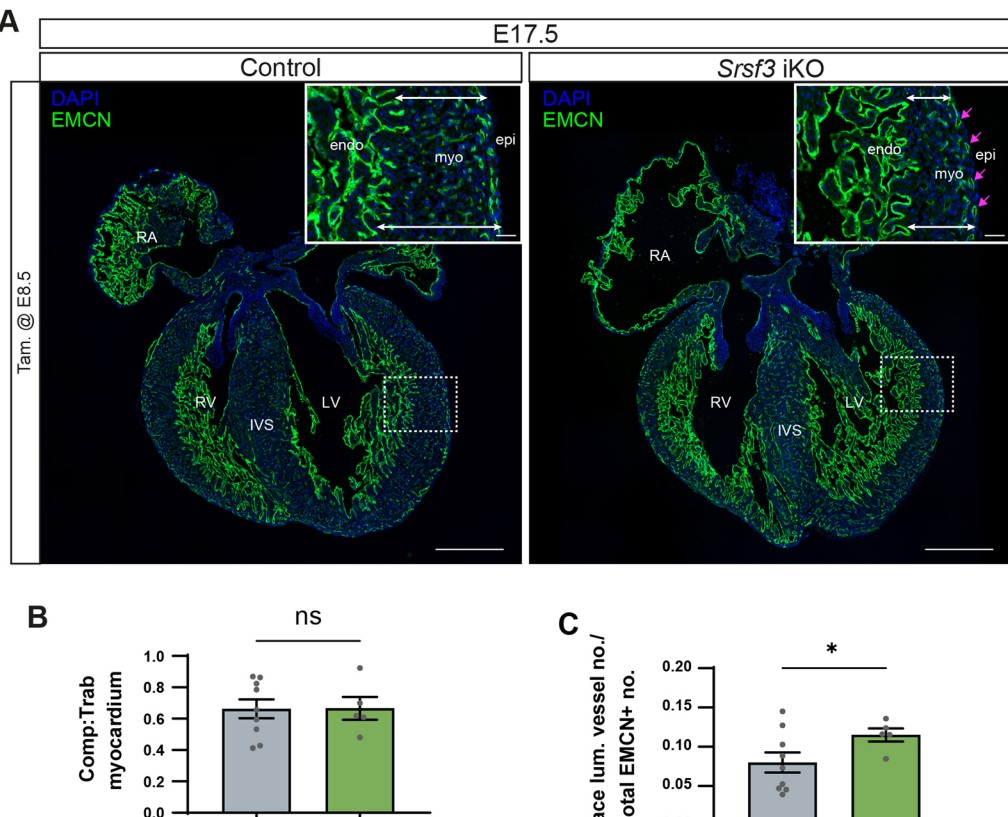

**Fig. 11. Defects in coronary vasculature persist with early-stage induction of SRSF3 depletion in Srsf3 iKO hearts.** (A) Immunofluorescence of the endocardium and vascular network (EMCN[+]) in control and *Srsf3* iKO embryonic mouse hearts at E17.5. Tamoxifen (Tam) was administered at E8.5. Insets show selected regions of interest (boxed) of the left ventricular wall. White arrows highlight myocardial wall thickness and magenta arrows highlight presence of superficial vessels. Scale bars: 500 μm (main panels); 50 μm (insets). Images representative of *n*=5-9 embryos. (B) Myocardial compaction, quantified as compact:trabeculated myocardial ratio. (C) Percentage surface lumenised vessels within 50 μm of the epicardium, as a percentage of the total number of lumenised vessels. Error bars indicate mean±s.e.m. (*n*=5-9 embryos). *P*<0.05 (unpaired *t*-test with Welch's correction). epi, epicardium; endo, endocardium; IVS, intraventricular septum; LV, left ventricle; myo, myocardium; ns, not significant; RV, right ventricle; RA, right atrium.

proliferative capacity in the absence of SRSF3. However, recombination in the Tg(Gata5-Cre) model also targeted a subset of cardiomyocytes. Thus, combined depletion of SRSF3 in most epicardial progenitor cells and a subset of cardiomyocytes contributed to the severe phenotype, since embryos with compromised epicardial formation generally die at later stages (Li et al., 2017; von Gise et al., 2011). Consistent with our findings, early-stage embryonic lethality was previously reported in cardiac-specific SRSF3 knockout mice due to impaired cardiomyocyte proliferation and survival (Ortiz-Sánchez et al., 2019).

Notably, peak epicardial SRSF3 expression at E11.5 coincides with the reported peak of epicardial proliferative activity (Wu et al., 2010). Temporal Cre-induction and selective deletion of *Srsf3* in epicardial cells, using the Wt1[CreERT2] line, resulted in impaired proliferation, ultimately leading to loss of SRSF3-depleted epicardial cells from the heart. SRSF3 was key to epicardial cell cycle progression as cyclin D1, found to be an SRSF3 target, was downregulated in SRSF3-depleted cells *in vitro*. Beyond regulation of a key cyclin, our irCLIP experiments mapped a significant proportion of SRSF3-RNA binding sites to a range of proliferation-associated transcripts in epicardial cells. To our knowledge, we present the first endogenous SRSF3 CLIP analysis, identifying RNA-binding targets without the confounding influence of protein tagging and overexpression. Importantly, this analysis validated SRSF3 as a positive regulator of G1-to-S phase transition in epicardium, consistent with roles demonstrated in other cell types (Kano et al., 2013; Kurokawa et al., 2013). A salient finding from our study was that the epicardial cells that escaped recombination and elimination could hyperproliferate to restore epicardial function; their elevated SRSF3 levels, compared with

control, reinforces the notion that SRSF3 drives cell cycle in the epicardium.

Of note, the role of SRSF3 in driving cancer progression has been investigated in the context of *MAPK4K* alternative splicing (Lin et al., 2018; Ajiro et al., 2016). In the most aggressive cancers, with the highest levels of *SRSF3*, high proliferation rates correlated with prominent expression of exon 16/17-containing *MAP4K4* isoforms; in contrast, cancers expressing lower levels of *SRSF3* proliferated more slowly and associated with *MAP4K4* isoforms in which these exons were predominantly skipped (Lin et al., 2018). *SRSF3* was directly implicated in the proliferation and invasiveness of U2OS and HeLa cancer cells, with *SRSF3* knockdown inhibiting cell proliferation, associated with *MAP4K4* exon skipping (Ajiro et al., 2016). We now demonstrate that SRSF3 engages similar mechanisms of alternative splicing to drive cellular proliferation during embryonic development.

SRSF3 depletion in the epicardium resulted in cell death associated with hypoxic stress, exemplified by upregulated *Fam162a* and an increase in TUNEL[+] epicardial cells. *Fam162a* is a transcriptional target of HIF1α, which promotes its expression under hypoxic conditions (Lee et al., 2004). Although we detected SRSF3 binding to *Fam162a* by irCLIP, we cannot distinguish the extent of direct regulation versus indirect mechanisms, which may be induced as a secondary consequence of hypoxic stress in *Srsf3* iKO hearts, due to vascular insufficiency as a result of inadequate epicardial stimulation of sinus venosus sprouting. It may be relevant that acute hypoxia in tilapia fish resulted in alternative splicing of *Fam162a* (Xia et al., 2018) and that de-regulated expression of RNA-splicing factors such as SRSF3 have been reported to mediate numerous alternative splicing events to promote cancer cell evasion

of apoptosis (Farina et al., 2020). Either way, SRSF3 depletion may expose epicardial cells to hypoxia-induced apoptosis, and the relationship between SRSF3 and *Fam162a* warrants further investigation.

Unlike targeting with Tg(Gata5-Cre), induction of SRSF3 deletion using Wt1[CreERT2] at E8.5 did not prevent epicardial formation, although it delayed formation. This could be due either to the reduced efficiency of the inducible Cre or the timing of Cre activation, occurring at the onset of proepicardial formation with Tg(Gata5-Cre), while the Wt1[CreERT2] suffers from a lag in recombination after tamoxifen delivery (Hayashi and McMahon, 2002). Although the epicardial layer was established in *Srsf3* iKO hearts, epicardial function was clearly defective, manifested as failed EMT, myocardial non-compaction and impaired coronary vessel formation, consistent with other studies (Martínez-Estrada et al., 2010; Smart et al., 2007; Tian et al., 2013; von Gise et al., 2011). In contrast, induction of SRSF3 depletion at E9.5 or E9.5/ E11.5 resulted in normal heart morphology at E17.5, due to the ability of non-recombined cells to overexpress SRSF3, hyperproliferate and restore the epicardial lineage. Thus, this study reveals an extraordinary potential of epicardial cells to compensate in the presence of genetic change, and highlights the limitations of variable recombination efficiency, which may confound investigation of crucial genes in the maturing epicardium. A significant number of studies use inducible Cre lines without considering the consequences of mosaic targeting, and accordingly risk misinterpreting data, especially when more readily recombined reporters such as tdTomato are used as surrogates for target gene recombination. The impact of incomplete recombination is compounded in the Wt1CreERT2 line, given the narrow developmental window for specific induction (Lupu et al., 2020b) and the potent mechanisms that drive enrichment of non-targeted cells. We highlight the shortcomings and key considerations exposed in our study to promote the appropriate use of inducible genetic models and the rigour and reliability of future investigations.

Our study adds to current literature describing the central role of SRSF3 in cellular proliferation (Ajiro et al., 2016; Änkö et al., 2010; Corbo et al., 2013; Gonçalves et al., 2009; Jia et al., 2010; Kano et al., 2013; Kurokawa et al., 2013). In the context of cardiac development, SRSF3 is required for the proliferation and survival of (pro)epicardial cells. Proper control of cell cycle progression is essential to enable epicardial cells to undergo EMT and support vascular development. This study defines SRSF3 as a key regulator of epicardial cell behaviour during development. Moreover, the delineation of the molecular mechanisms controlling epicardial proliferation may provide relevant insights for regenerative therapies.

## MATERIALS AND METHODS
### Mouse strains
Conditional and inducible targeting of SRSF3 was achieved by crossing the epicardial Cre lines Tg(Gata5-Cre)1Krc (Merki et al., 2005) and Wt1[tm2(cre/ERT2)Wtp] (Zhou et al., 2008), respectively, with mice in which exons 2 and 3 of *Srsf3* were flanked by loxP sites (Jumaa et al., 1999; Ortiz-Sánchez et al., 2019). The epicardial lineage was traced using Rosa26tdTomato in crosses with Gt(ROSA)26Sor[tm14(CAG-tdTomato)Hze] (Madisen et al., 2010). Controls used were either Cre+; *Srsf3*[+/+] or littermate *Srsf3*[fl/+] (tamoxifen induced, in the case of iKO), or *Srsf3*[fl/fl]; Cre−. Pregnant females were administered 40 mg/kg at E9.5 and E11.5, or 80 mg/kg tamoxifen at E8.5 or E9.5 (where specifically stated), by oral gavage. All procedures were approved by the University of Oxford Animal Welfare and Ethical Review Board, in accordance with Animals (Scientific Procedures) Act 1986 (Home Office, UK).

### Tissue harvesting and cryosectioning
Embryos and embryonic hearts were harvested and fixed in 4% paraformaldehyde for 2 h at room temperature (RT). Following PBS washes, tissues were equilibrated overnight at 4°C in 30% sucrose in PBS, and then gradually transitioned into O.C.T. Samples were stored at −80°C, then cryosectioned at 8-12 μm thickness.

### (Pro)epicardial explant and cell line culture
Embryonic hearts were dissected into atrial and ventricular pieces and cultured on 1% gelatine-coated dishes in culture medium [15% fetal bovine serum (FBS), 1% Penicillin/Streptomycin, DMEM high glucose GlutaMAX]. Explants were incubated at 37°C in 5% $CO_2$ for 3 days. Dissected proepicardia were subjected to the same culture conditions as above. The strategy to induce SRSF3 depletion *in vitro* involved culturing epicardial explants in culture medium supplemented with 1 μM 4-OHT. An immortalised EPDC line (Austin et al., 2008) was cultured on 1% gelatine in immorto medium (10% FBS, 1% Penicillin/Streptomycin, ITS supplement, 0.1% IFNγ, DMEM high glucose GlutaMAX) at 33°C in 5% $CO_2$. The MEC1 epicardial cell line (Li et al., 2011) was obtained from Sigma-Aldrich and cultured on 0.1% gelatine in DMEM high glucose with GlutaMAX, 10% FBS, 1% Penicillin/Streptomycin, at 37°C in 5% $CO_2$. Cells were transfected with 40 nM or 80 nM siRNA (siControl: silencer siRNA negative control, 4390843; siSrsf3 #1: s73613; siSrsf3 #2: s73615; Thermo Fisher Scientific) for 48 h using Lipofectamine RNAiMAX Transfection Procedure (Thermo Fisher Scientific, 13778-100).

### Immunofluorescence
Cryosections were rehydrated, permeabilised with 0.5% Triton X-100 in PBS for 15 min, then incubated for 1 h at RT in blocking buffer [1% bovine serum albumin and 10% donkey serum, in 0.1% Triton X-100 in PBS (PBST)]. Primary antibodies (Table S24) were added, and sections incubated overnight at 4°C. Sections were washed with PBST (three 10 min washes), then secondary antibodies were added, and incubated for 1 h at RT, followed by more PBST washes, as above. DAPI (4′,6-diamidino-2-phenylindole) was used to stain nuclei. Cultured epicardial explants, FACS-sorted epicardial lineage and epicardial cell line (Austin et al., 2008) were subjected to the same immunofluorescence protocol, but without the initial permeabilisation (0.5% Triton X-100) step. Images were acquired using a Leica DM6000 fluorescence microscope or Olympus Fluoview-1000 confocal microscope and processed using Fiji software (Schindelin et al., 2012). Quantification was carried out using Fiji or CellProfiler (McQuin et al., 2018) software.

### Quantitative assessment of surface lumenised coronary vessels
Whole-heart images were processed using Fiji software (Schindelin et al., 2012). A mask was applied based on DAPI (nuclei) staining and, through a series of dilations and shrinking, the whole cardiac tissue was segmented (region of interest 1; ROI 1). A region of 30-50 μm in depth from the epicardial border was automatically selected by segmenting 3% of the total ventricular area (outer region; ROI 2). Particle analysis was used to quantify the number, area, mean intensity, and perimeter of vessels and nuclei within the ROI 2. Particle analysis was used to quantify the area of the myocardium below the epicardium (ROI 2) and total ventricular area (ROI 1).

### Quantitative assessment of hypoxic regions in the heart
Whole-heart images were processed using Fiji software (Schindelin et al., 2012). A mask was applied based on cardiac troponin staining (myocardium) and, through a series of dilations and erosions, the whole myocardium was segmented. The atria and highly trabeculated endocardial region were removed, to leave mostly compact ventricular myocardium (region of interest 1; ROI 1). Hypoxic regions with high GLUT1 levels within ROI 1 were segmented by setting a fluorescence intensity threshold (ROI 2). Particle analysis was used to quantify the total area of compact myocardium (ROI 1) and hypoxic regions (ROI 2).

### RNA extraction and qRT-PCR
RNA was extracted from embryonic hearts using RNeasy Mini kit (QIAGEN) according to the manufacturer's instructions. cDNA was prepared with High-Capacity cDNA Reverse Transcription kit (Thermo

Fisher Scientific) according to the manufacturer's instructions and analysed by quantitative RT-PCR using the Fast SYBR Green Master Mix (Thermo Fisher Scientific). Primers are listed in Table S25.

## Flow cytometry
For quantifying the epicardial lineage, E15.5 or E17.5 whole hearts were dissociated, and surface-stained with viability dye, then pre-treated for 5 min on ice with TruStain FcX antibody to block Fcγ receptors. Cells were subsequently stained with BV605 rat anti-CD31 antibody, on ice for 0.5 h. Cells were fixed and permeabilised, then intracellularly stained with cTNT as previously described (Lupu et al., 2020b; Redpath et al., 2021). Cells were stored in IC fixation buffer (Thermo Fisher Scientific, 00-8222-49) until acquisition.

## scRNA-seq and analysis
For E13.5 scRNA-seq data, whole hearts from E13.5 embryos were dissociated as described previously (Lupu et al., 2020b; Redpath et al., 2021), then processed using the Chromium Single Cell 3′ Reagent Kit (v3 chemistry) (10x Genomics, PN-1000092) as indicated by the manufacturer's instructions. The samples were sequenced in three rounds using the NextSeq® 500/550 High Output Kit v2 (150 cycles) 400 million reads (8000 cells) (Illumina, FC-404-200) on the Illumina NextSeq 500 platform. The reads were processed using CellRanger (10x Genomics), then analysed in Seurat 3 (Stuart et al., 2019) as described below. Cells with more than 20% mitochondrial reads and more than 7000 genes were removed from the analysis. Uniform manifold approximation and projection (UMAP) was used to visualise samples. Cell cycle S- and G2M-phase scores assigned for each individual cell (Seurat) were plotted as a scatter graph. For E13.5, three control hearts and two *Srsf3* iKO hearts were used.

For the E15.5 scRNA-seq data, the ventricles from E15.5 embryonic hearts were dissociated as described previously (Lupu et al., 2020b; Redpath et al., 2021), and BD FACSAria III was used to sort live cells positive for tdTomato fluorescence into 96-well plates. The cells were processed according to the Smart-Seq2 protocol (Picelli et al., 2014). Tagmentation and library preparation was carried out using the Nextera XT DNA Library Prep kit and sequenced using the NextSeq 500/550 High Output Kit v2 (75 cycles) 400 million reads (Illumina, FC-404-2005) on the Illumina NextSeq 500 platform. BCL files from the sequencer were converted to FastQ with bcl2fastq (Illumina, v2.19.1.403) using default settings. Samples with fewer reads in total than the empty wells were discarded. Reads were trimmed using default settings and '—nextera' for adapter clipping (TrimGalore v0.4.4). Random samples were checked with FastQC (v0.11.3) to control for the quality of the sequence. Reads were aligned to the mouse genome (GRCm38/Mm10) with STAR (v.2.3.3a) in gene counting mode with Gencode (vM16) annotations minus the megatranscript Gm20388. Splice junctions from all samples were combined after the first pass alignment. Non-canonical junctions and junctions covered with fewer than ten reads were removed prior to running the second pass alignment. The samples were further processed with Seurat. Genes with >200 reads in at least three cells were retained for downstream processing. Cells were filtered to retain those with <5% mitochondrial genes and at least 500 expressed genes. Reads were normalised to sequencing depth and scaled. For E15.5, six control hearts and four *Srsf3* iKO hearts were used.

To detect and quantify deletion of exon 2+3 from single cells (using scRNA-seq), analyses of sequencing data were performed using custom Python 3 scripts utilising the pysam (v0.22.0) library and visualised in IGV (v2.16.2) as Sashimi plots. First, an unbiased discovery of all splice junctions within the *Srsf3* genomic locus was performed. A custom Python script was used to scan all reads within the target region corresponding to the *Srsf3* locus (chr17:29032000-29042500) in the input BAM file (10x Genomics data). For each read, alignment blocks were inspected using pysam's get_blocks() method. Any gaps between contiguous blocks greater than 200 bp were considered potential splice junctions. The start and end coordinates of each unique junction were recorded, and the total number of reads supporting each junction was counted. This approach allowed for the initial identification of both the canonical wild-type splice junctions and novel, deletion-specific junctions. Following discovery, a targeted quantification and zygosity analysis was performed, and junctions mapped to each individually barcoded cell. Cell annotations were leveraged to report

the count and percentage of cells in each genetic state, grouped by cell type. Custom scripts for single cell splicing analysis have been made available at https://github.com/loganminhdang/singlecellvariantanalysis.

## MEC1 epicardial cell bulk RNA-seq (with longer reads for splicing analysis)
mRNA was extracted from epithelial monolayer cultures using an RNeasy Plus Micro kit (QIAGEN), according to the manufacturer's instructions. Library preparation and sequencing (NovoSeq X, 50 million reads per sample, paired end 350 bp reads) were performed by GENEWIZ.

RNAseq data were analysed with the nf-core RNAseq pipeline v3.18.0 (10.5281/zenodo.1400710). Reads were aligned to the mm39 genome using the Star Salmon aligner. Data were further analysed with DESeq2 (Love et al., 2014) in R studio (Posit team, 2025; RStudio: Integrated Development Environment for R. Posit Software, PBC, Boston, MA; http://www.posit.co/). Differentially expressed genes were annotated with AnnotationHub (Morgan, M. and Shepherd, L., 2025; AnnotationHub: Client to access AnnotationHub resources; doi:10.18129/B9.bioc.AnnotationHub).

## Splicing analysis
Splicing analysis was conducted on the RNA-seq data using the rMATS-turbo package (Wang et al., 2024). All samples were processed together in the pre-steps. The post-step was run separately using the flags -t paired — statoff. Comparison specific information was extracted with the prepare_stat_inputs.py tool, before running the statistical model separately on pairs of groups. Output from the different comparisons were combined in R studio and GO term analysis of alternatively spliced genes performed with clusterProfiler (Wu et al., 2021), removing redundant terms with the simplify command.

## Alternative polyadenylation analysis
Alternative polyadenylation analysis was performed on the RNA-seq data using QAPA (Ha et al., 2018). Since no pre-calculated 3′ UTR library was available for mm39, the 3′ UTR library was built from annotation with QAPA-build. We bypassed the poly(A) annotation step by include the -N option, since no PolyA annotation file was available for mm39 and used a *genepred file generate from gencode.vM37.primary_assembly.annotation.gtf with UCSC-gtftogenepred tool. The 3′ UTR sequences were extracted from the mm39 genome using the qapa fasta command with the —decoys flag. The 3′ UTR library was subsequently indexed with Salmon-index tool (Patro et al., 2017) and the generated indexes then used to quantify 3′UTR counts with salmon-quant tool and fastq files as input. Finally, 3′UTR usage was quantified with the QAPA-quant tool and significant events were identified in R studio by filtering the data and running *t*-tests on siCtrl versus siSrsf3 samples. Significant events were defined as those with *P*-values below 0.05, changes in polyadenylation usage of at least 20% and expression values higher than 1 TPM.

## Epicardial *Srsf3* irCLIP and analysis
irCLIP was performed using the epicardial cell line, largely as described previously (Ewels et al., 2020; Kolberg et al., 2020; Martin, 2011; Robinson et al., 2011; RStudio Team, 2015; Zarnegar et al., 2016) with some adaptations of the protocol. Primers were ordered as published from IDT. The L3-biotin pre-adenylated linker was ordered with the IRdye-800CW attached from IDT: /5rApp/AGATCGGAAGAGCGGTTCAGAAAAAAAAAAAA/iIRD-800CWN/AAAAAAAAAAAA/3Bio/. Cells were washed with PBS and crosslinked with 200 mJ/cm² UV light in a Stratalinker 1800. Protein A Dynabeads (Life Technologies) were washed and coated with 10 µg anti-SRSF3 antibody (ab198291, Abcam). Cells were resuspended in lysis buffer [50 mM Tris-HCl, pH 7.4, 100 mM NaCl, 1% Igepal CA-630, 0.1% SDS, 0.5% sodium deoxycholate, protease inhibitor (Sigma Aldrich)] and samples sonicated in a Bioruptor (Diagenode). In the presence of Turbo DNase (Thermo Fisher Scientific), RNA in 100 µg lysate per sample was partially digested with 1/500 RNase I or 1/20 RNaseI (Invitrogen) for the high RNase I control sample. SRSF3 and crosslinked RNA were immunoprecipitated from lysate for 2 h at 4°C with rotation. Beads were washed once each with high stringency buffer (20 mM Tris, pH 7.5, 120 mM NaCl, 25 mM KCl, 5 mM EDTA, 1% Triton X-100, 1% sodium deoxycholate), high salt buffer (20 mM

Tris, pH 7.5, 1 M NaCl, 5 mM EDTA, 1% Triton X-100, 1% sodium deoxycholate, 0.001% SDS) and low salt buffer (20 mM Tris, pH 7.5, 5 mM EDTA). RNA was dephosphorylated with polynucleotide kinase (NEB) at 37°C for 30 min, followed by washes with high salt buffer and NT2 buffer (50 mM Tris, pH 7.5, 150 mM NaCl, 1 mM MgCl$_2$, 0.0005% Igepal). The L3-App-IRD800CWN-biotin linker was ligated to the RNA with T4 RNA ligase (NEB) overnight at 16°C. Beads were washed with high salt and NT2 buffer and eluted in NuPAGE loading buffer (Invitrogen) and run on a NuPAGE 4-12% Bis-Tris gel (Invitrogen) and transferred onto a 40 µm nitrocellulose membrane. The SRSF3-RNA complexes were visualised with an Odyssey Clx infrared imager. Protein-RNA complexes containing RNA of approximately 70-280 nt length (20-80 kDa above the protein-RNA-adapter complex) were excised and RNA released by proteinase K (Roche) digestion for 60 min at 50°C in proteinase K digestion buffer (100 mM Tris, pH 7.5, 50 mM NaCl, 1 mM EDTA, 0.2% SDS). RNA was extracted with phenol-chloroform (Sigma-Aldrich) and precipitated with ethanol overnight. RNA was reverse transcribed with TIGRT-III reverse transcriptase (Ingex) and cDNA/RNA hybrids captured with MyOne Streptavidin C1 Dynabeads (Life Technologies). RNA was degraded by alkaline hydrolysis for 15 min in RNA degradation buffer (0.1 µM P3short oligonucleotide, 3.75 mM MnCl$_2$) followed by cDNA circularisation with Circligase II (Epicentre). Circularised cDNA was captured on Ampure XP beads (Beckman Coulter) before elution in water. The cDNA was amplified with Phusion HF polymerase (NEB) and P3/P6 primers in a ViiA7 qPCR machine in the presence of SYBR Green (Thermo Fisher Scientific) for real-time visualisation of the amplification product. The library was cleaned-up with Select-a-Size DNA Clean & Concentrator kit (Zymo Research) to remove empty library before a further three PCR cycles using P3/P6Solexa primers to add the adapters necessary for Illumina sequencing. The final library was cleaned up first with Ampure XP beads followed by gel clean-up on a Novex TBE-6% Urea gel. Libraries were visualised with SYBR Gold (Thermo Fisher Scientific) and excised to remove residual empty library. DNA was extracted from the gel in Crush-Soak Gel buffer (500 mM NaCl, 1 mM EDTA, 10 mM Tris pH 7.5, 0.1% SDS) overnight at 55°C and cleaned up using DNA Clean-and-Concentrator-5 kit (Zymo Research). Libraries were quantified with KAPA Library Quantification Kit (Roche) and sequenced on a NextSeq 550 with a High Output Kit v2 (75 Cycles) (Illumina).

Sequencing data were demultiplexed with Cutadapt 2.1 (Martin, 2011) and processed with the nf-core/clipseq pipeline (Ewels et al., 2020). Crosslinks from the three replicates were merged and used as input to iCount-Mini (Robinson et al., 2011) using an FDR threshold of <0.05 as significant. Peaks less than 5 nt wide were filtered out. For motif analysis, we examined the top 20 pentamers occurring in the −20…+20 window around the peak starts. GO term enrichment analysis was performed using gprofiler2 (v0.2.3) (Kolberg et al., 2020) using genes with greater than ten peak crosslinks and a background set defined as all genes with crosslinks. Reactome analysis was performed using ReactomePA package (v1.46.0) (Yu and He, 2016). IGV viewer (Robinson et al., 2011) was utilised to browse irCLIP data and clipplotr (Chakrabarti et al., 2023) was utilised for irCLIP data visualisation.

### Whole-mount DAB staining
Embryonic hearts were placed in 4% paraformaldehyde in 1.5 ml tubes after dissection, then left overnight at 4°C on a rotator. In between each of the following steps, hearts were washed three times for 15 min in PBST. Hearts were incubated for 1 h at RT in blocking buffer (5% goat serum in PBST). An antibody against PECAM1, 1:500 dilution in PBST with 1% bovine serum albumin, was added and incubated overnight at 4°C. Hearts were incubated in biotinylated secondary antibody (goat anti-Armenian hamster Biotin, Abcam, ab5744) at 1:250 dilution in PBST for 1 h. Avidin-biotin horseradish peroxidase complex (ABC Elite, Vector Laboratories, PK-4000) at 1:50 in PBS was added for 30 min. Hearts were placed in DAB substrate (Vector Laboratories peroxidase substrate kit, SK-4100). When vessels were visible, the reaction was stopped by rinsing the hearts with ddH2O. The hearts were imaged on a stereo microscope.

### Western blotting
Protein was extracted using RIPA buffer (50 mM Tris-HCl pH 7.4, 1% Triton X-100, 0.1% SDS, 150 mM NaCl, 2 mM EDTA) with protease inhibitors (Sigma-Aldrich, P8340) and phosphatase inhibitors (Sigma-Aldrich, P5726). Protein concentration was quantified using Pierce BCA Protein Assay kit (Thermo Fisher Scientific, 23227), then 6 µg of protein was loaded on a 4-20% MiniProtean Gel (Bio-Rad, 4561096). The samples were transferred using the Trans-Blot Turbo Transfer Pack (Bio-Rad, 1704456), blocked in 8% skimmed milk powder in TBST (20 mM Tris, pH 7.5, 150 mM NaCl, 0.1% Tween 20) for 1 h, then incubated in 1:1000 SRSF3 (Abcam, ab73891) and 1:5000 GAPDH (Abcam, ab8245) antibodies in blocking buffer (5% skimmed milk powder in TBST) overnight at 4°C, followed by three washes of 5 min each in TBST and incubation with HRP-conjugated secondary antibodies, diluted 1:2500 in blocking buffer. The blot was rinsed three times for 5 min each wash in TBST, then the samples were detected using ECL™ Western Blotting Detection Reagent (Sigma-Aldrich, GERPN2209).

### Fluorescence *in situ* hybridisation (mRNA)
RNAscope Multiplex Fluorescent v2 assay (ACD) was performed on cryosections according to manufacturer's instructions, with minor modifications as previously described (Lupu et al., 2020b). Ndrg1 and negative control probe, and TSA plus fluorophores, are listed in Table S26.

### TUNEL assay
Click-iT™ Plus TUNEL assay (Thermo Fisher Scientific) was performed on cryosections according to the manufacturer's instructions.

### *Ccnd1* and *Ccng1* mRNA stability assay following transcription inhibition by actinomycin D
mRNA stability was assayed in MEC1 cells 48 h after transfection with 40 nM siRNA (siControl: silencer siRNA negative control, 4390843; siSrsf3 #1: s73613; siSrsf3 #2: s73615; Thermo Fisher Scientific) using Lipofectamine RNAiMAX (Thermo Fisher Scientific, 13778-100), essentially as previously described (Ratnadiwakara and Änkö, 2018). Cells were harvested over a 36-h period, following treatment with 5 µg/ml actinomycin D (Merck). mRNA was extracted using the Monarch(R) Spin RNA Mini Isolation Kit. *Ccnd1* and *Ccng1* transcripts were measured by qRT-PCR, as described above, normalised against *Msln*, to determine half lives and decay rate constants.

### Statistical analysis
Statistical analysis was performed in GraphPad Prism 9 software. Unpaired *t*-test with Welch's correction was used to compare two experimental groups. One-way Welch ANOVA and Dunnett's multiple comparison test was used to compare more than two experimental groups. A *P*-value lower than 0.05 was considered significant. Details of statistical analyses, including biological replicate numbers, are included in figure legends.

### Acknowledgements
We thank Prof. Peter Nielsen, Max Planck Institute, and Prof Enrique Lara-Pezzi, CNIC, for the *Srsf3*$^{fl/fl}$ line; Prof. William Pu, Harvard University, for the Wt1$^{CreERT2}$ line; Prof. Kenneth Chien, Karolinska Institutet, for the Tg(Gata5-Cre) line; Dr Madeleine Lemieux (Bioinfo) for processing SMART-Seq2 and 10x data; Robert Hedley and Michal Maj for providing technical assistance for FACS-sorting epicardial cells at The Don Mason Facility of Flow Cytometry, Sir William Dunn School of Pathology, University of Oxford; Paul Sopp for providing technical assistance for FACS-sorting epicardial cells for Smart-seq2 at the WIMM Flow Cytometry Facility, MRC Weatherall Institute of Molecular Medicine, University of Oxford; and Neil Ashley for providing technical assistance and services at the MRC WIMM Advanced Single Cell OMICS Facility (WASCOF), MRC Weatherall Institute of Molecular Medicine, University of Oxford.

### Competing interests
The authors declare no competing or financial interests.

### Author contributions
Conceptualization: I.-E.L., N.S.; Data curation: I.-E.L., S.B., A.M.C.; Formal analysis: I.-E.L., S.B., A.M.C., I.R.M., Q.M.D., T.C., A.N.R., N.S.; Funding acquisition: S.D.V., N.S.; Investigation: I.-E.L., S.B., Q.M.D., A.N.R., N.S.; Methodology: I.-E.L., S.B., A.M.C., I.R.M., A.N.R.; Project administration: N.S.; Resources: N.S.; Supervision: A.M.C., A.N.R., N.S.; Writing – original draft:

I.-E.L., A.N.R., N.S.; Writing – review & editing: I.-E.L., S.B., A.M.C., I.R.M., A.N.R., N.S.

## Funding
This work was funded by the British Heart Foundation (BHF): DPhil Studentship (FS/15/68/32042); BHF Project grants (PG/15/112/31940 and PG/24/11910); BHF Ian Fleming Senior Basic Science Research Fellowships (FS/13/4/30045 and FS/19/32/34376); BHF Centre of Regenerative Medicine, Oxbridge (RM/13/3/30159). Open Access funding provided by the University of Oxford. Deposited in PMC for immediate release.

## Data and resource availability
All sequencing data from this study are available in Gene Expression Omnibus database under accession numbers GSE145832, GSE205797 and GSE313583, or ArrayExpress under accession number E-MTAB-11853. Custom scripts for single-cell splicing analysis are available at https://github.com/loganminhdang/singlecellvariantanalysis. All other relevant data and details of resources can be found within the article and its supplementary information.

## Peer review history
The peer review history is available online at https://journals.biologists.com/dev/lookup/doi/10.1242/dev.204918.reviewer-comments.pdf

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
