## [Peer Review File · Development (Cambridge, England)]

The RNA-binding protein SRSF3 controls epicardial formation by regulating splicing and proliferation

Irina-Elena Lupu, Susann Bruche, Anob M. Chakrabarti, Ian R. McCracken, Quang M. Dang, Tamara Carsana, Sarah De Val, Andia N. Redpath and Nicola Smart
DOI: 10.1242/dev.204918

Editor: Benoit Bruneau

Review timeline

Submission to Review Commons:	17 January 2025
Submission to Development:	6 May 2025
Editorial Decision:	20 May 2025
First revision received:	5 January 2026
Accepted:	18 January 2026

Reviewer 1

Evidence, reproducibility and clarity

Summary:

The study by Irina Lupu and colleagues highlights SRSF3 as a key regulator of epicardial development by regulating epicardial cell proliferation. This was demonstrated via two murine knockout models; the first to assimilate the role SRSF3 plays in epicardial formation as a whole, and the second to address its importance post a pivotal maturation point. Through scRNA sequencing and iCLIP, several SRSF3 targets were ascertained and identified as cell cycle regulators. Those epicardial cells that did not lose SRSF3 compensated the loss of some of their mates by increasing SRSF3 expression and over- proliferating. Overall, the paper is interesting and the conclusions are largely supported by the provided data.

Major comments:

1. Authors claim that a "reduction in SRSF3 expression levels coincided with the downregulation of WT1 in the epicardium". This was evidenced by immunofluorescence imaging (figure 1A). I suggest conducting a qRT-PCR to quantify Wt1 expression over time, similar to the experiment they performed in figure 1B.
2. A western blot depicting SRSF3 protein production in controls compared to the knockout model may provide stronger evidence of its depletion (figure 1E).
3. Authors state that they were unable to directly identify the absence of exons 2 and 3 in individual cells. Please provide evidence that exons 2 and 3 have been knocked out, at least by performing a qRT-PCR.
4. To prove the functional implication in the observed phenotype of the identified SRSF3 targets, please interfere with Map4k4 activity or expression and check whether the defective epicardial cell proliferation is reverted. This should be done at least in vitro, ideally in vivo.

Minor comments:

1. Several minor typos and spacing issues were observed. Please correct.
2. It would be good for the reader if the authors would simplify their rationale for the use of the two mouse models. It is slightly convoluted and not easy to follow.
3. In figure 4, it is recommended to add a stacked bar plot to represent the percentage of each

cell cluster/population after 4A. This would help the reader

4. Figure 4B. It is confusing for the reader to understand the fact that the majority of tdTomato+ sorted cells in Srsf3 iKO keep expressing Srsf3. Including the quantification of the image could help.

Significance

The paper will be of interest to readers in the field of cardiology, embryology and molecular biology. It will advance the field especially in the study of the development of the epicardium. The models are sophisticated and the experiments carefully performed.

My field is molecular cardiology, with interest in RNA-binding proteins.

Reviewer 2

Evidence, reproducibility and clarity

In this study, Lupu et al. analyzed the role of the RNA-binding protein SRSF3 for epicardial development. The authors found that Srsf3 is highly expressed in the proepicardial organ and during early stages of epicardial layer formation. Conditional inactivation of SRSF3 in the proepicardial organ stage using a Gata5-Cre driver line resulted in defective formation of the epicardium, accompanied by a proliferation arrest of the proepicardium, resulting in embryonic lethality at E12.5. In contrast, epicardial-specific Srsf3 deletion at later stages using the inducible Wt1CreERT2 line caused a less severe phenotype indicated by impaired coronary vasculature formation, reduced cardiac compaction, and myocardial hypoxia. Mosaic recombination yielded a small population of epicardial cells that upregulate Srsf3, hyperproliferate and compensate for the depleted Srsf3 negative lineage. Single-cell RNA sequencing of control and epicardial Srsf3 knock out hearts, combined with infrared CLIP to map SRSF3 binding sites in the transcriptome identified a number of putative SRSF3 targets involved in mitotic cell cycle control. Among others, SRSF3 binds directly to transcripts encoding key regulators of proliferation, such as Cyclin D1, and senescence, including MAP4K4. The authors conclude that SRSF3 exerts different functions in processing of RNAs, including splicing.

Overall, this is a well-written and well-organized manuscript, describing interesting findings in the field of epicardial development. However, the mechanistic part is not overly strong. The authors detected some moderate changes in the distribution of different Map4k4 splicing isoforms after knockdown of Srsf3 in an immortalized epicardial cell line but did not go any deeper. The cause for the reduced presence of transcripts for the SRSF3-target Ccnd1 after knockdown of Srsf3 remains enigmatic.

Significance

The authors raise a number of speculations why remaining Srsf3-expressing cell start to hyperproliferate after inactivation of Srsf3 but it does not become clear which mechanism is critical. How do non-targeted epicardial cells in the mosaic recombination model sense the loss of SRSF3 knock out cells, resulting in hyperproliferation and enhanced Srsf3 expression? Is a loss of lateral inhibition, e.g. by activated YAP/TAZ, causative for enhanced proliferation of the remaining epicardial cells and an elevated expression level of WT1 and SRSF3? Immunofluorescence staining and/or qRT-PCR of YAP/TAZ and TEADs might provide an answer.

Is the elevated expression of Srsf3 in non-targeted epicardial cells due to enhanced transcription and/or by altered post-transcriptional processes? How does this observation fit to previous reports indicating that Srsf3 overexpression promotes inclusion of an autoregulatory cassette exon (exon 4) containing a premature (in-frame) stop codon in Srsf3, thereby confining this SRSF3 isoform to nonsense mediated decay (NMD) (doi: 10.1093/emboj/16.16.5077, doi: 10.1186/gb-2012-13-3-r17, doi: 10.1038/srep14548, doi: 10.1161/CIRCRESAHA.118.31451)? In contrast, Srsf1 as well as PTBP1/2 have been previously reported to regulate Srsf3 expression by promoting exon 4 skipping. The authors should perform RNA seq and/or qRT-PCRs validation to check the inclusion of Srsf4

Exon4 as well as Srsf1 and PTBP1/2 expression levels in control and knock out epicardial cells.

It remains unclear by which mechanisms (alternative splicing, alternative polyadenylation, miRNA processing, or others) SRSF3 mainly exerts its function in the embryonic epicardial lineage. The selection and validation of Map4k4 as a splicing target is not based on an unbiased splicing analysis. In my opinion it is mandatory to provide a full assessment of splicing changes in Srsf3-deficient cells, either by long-range sequencing or by analysis of exon-junction reads.

Likewise, it is completely enigmatic what SRSF3 does to Ccnd1 transcripts. Does SRSF3 increase the half-life of Ccnd1, does it impact trafficking? At least, the authors have to determine changes in the half-life of Ccnd1 after depletion of SRSF3.

An unbiased bioinformatics analysis addressing alternative splicing, alternative polyadenylation, and mRNA processing is necessary. Ideally, primary epicardial cells should be used and not an immortalized epicardial cell lines. It is well known, that splicing in cell lines differs substantially from splicing in primary cells.

I am not convinced that the moderate changes of different Map4k4 splicing isoforms after knockdown of Srsf3 are really responsible for the rather drastic phenotype. Additional experiments are needed to prove a decisive function of a shift in Map4k4 splicing isoforms for hyperproliferation of epicardial cells.

The authors claim that inactivation of Srsf3 inhibits cell proliferation and causes a senescence-like phenotype. The claim for acquisition of senescence is solely based on transcriptional changes. No attempts were made to visualize an increase of senescent cells in Srsf3-mutant embryos. The authors need to perform SA-βGAL assays or use other techniques to analyse the appearance of senescent cells in the mutants.

Fig. 2E indicates that WT1 positive / tdTom negative epicardial cell population is enriched in a specific region of the pre-epicardial organ from Srsf3KOs. However it is not clear whether these cells proliferate. The authors should quantify Ki67positive cells in both the WT1- positive / tdTom-positive and the WT1positive / tdTom-negative epicardial population.

In the headline on page 6, the authors stated that "SRSF3 depletion in the PEO results in impaired ... migration of epicardial progenitor cells", which they deduced from the reduced outgrowth of ventricular epicardial explants. However, the reduced outgrowth from the PEO could be caused by both, reduced proliferation and/or reduced migration. Therefore, the authors should provide additional data clearly indicating reduced migration, e.g. by blocking transcription. Scratch assays of SRSF3 knockout/knockdown vs. control epicardial cells would strengthen the analysis, Is there a change in the GO term "regulation of migration"?

To prove the reduced proliferation ratio in Figure 4B, quantification of Cyclin D1 positive cells in both SRSF3 positive and negative cells is required.

Minor issues

Abstract line 12: a full stop is missing at the end of the sentence.

Figure 1A: E11.5, figure label 'DAPI WT1' is missing.

Page 8: Bracket in front of Fig. 4B is missing

Page 8: G2M phase change uniformly to G2/M phase

Page 9: 'Srsf3-depleted hearts also demonstrated an increased abundance of epicardial cells with upregulated expression of genes associated with quiescence, such as Clu48, and senescence, for example Map4k4, Tmem30a and Pofut249 (Fig. 4F)'. The sentence is misleading, implying Srsf3 inactivation in all cardiac cell types ('Srsf3-depleted hearts').

Reviewer 3**Evidence, reproducibility and clarity**

Lupu et al have identified a role for the RNA binding protein SRSF3 in epicardial development. In a well written manuscript containing many experiments the authors show that this protein is required at different time points in epicardial development to control a range of processes, in particular cell proliferation. This advances understanding of the complex roles of this RNA binding protein in the heart - and raises an important message about how incomplete Cre recombination needs to be considered in interpreting conditional mutant phenotypes. The following points should be addressed.

1. SRSF3 is known to play essential developmental roles in the myocardium where it regulates capping of transcripts involved in contraction. This point should be mentioned in addition to roles in proliferation. To facilitate understanding, the authors should say more about the subset of cardiomyocytes labelled by Gata5-Cre. For example, is this the result of stochastic activation of the transgene or is a specific subset of cells labelled? How much of the myocardium is targeted?
2. The authors show failure of ventricular compaction at E13.5 using Wt1-CreERT2 and go on to assess proliferation in epicardial cells. As epicardial-derived signals are known to promote compact myocardial growth, they should also show whether there are indirect defects in proliferation in compact layer myocardium that might explain the non-compaction phenotype. The authors should also indicate if any of the large number of genes bound by SRSF3 encode known or potential pro-proliferative signals from the epicardium or EPDCs to the myocardium and potentially validate their altered expression in mutant hearts.
3. The rescue by expansion of non-recombined cells is a most interesting aspect of this study. Can the authors see any such outcompeting in the explant experiments (for example in Figure 2)? Do the authors consider this to be an exclusively in vivo competition phenomenon? Given the known roles of Myc in cell competition can the authors use their single cell transcriptomic data to score Myc expression levels in cells from Srsf3 iKO hearts or determine if Myc transcripts are bound by SRSF3?
4. The authors suggest that this rescue occur by upregulation of Srsf3 in non-recombined cells. It would be helpful to provide additional lines of evidence supporting the hypothesis that SRSF3 expression is upregulated due to hypoxia. Do the CLIP experiments reveal whether SRSF3 binds to its own transcript?
5. The authors imply that SRSF3 may regulate Ccnd1 mRNA stability. Can the authors directly evaluate this point? Please clarify if this gene is also affected in the knock-down experiments in MEC1 cells.
6. Please brighten the immunofluorescence panels in Figure 1 to more clearly show nuclear labelling and tissue structure.
7. Given the broad roles of SRSF3 is the adjective key necessary in the title?

Significance

This ms advances understanding of the complex roles of this RNA binding protein in the heart - and raises an important message about how incomplete Cre recombination needs to be considered in interpreting conditional mutant phenotypes. This study would be of interest to researchers in the fields of heart development and RNA-protein interactions. Although there are a number of major points to be addressed, these could be potentially dealt with rapidly.

Author response to reviewers' comments

Manuscript number: RC-2025-02875

Corresponding author(s): Nicola, Smart

[The "revision plan" should delineate the revisions that authors intend to carry out in response to the points raised by the referees. It also provides the authors with the opportunity to explain their view of the paper and of the referee reports.]

The document is important for the editors of affiliate journals when they make a first decision on the transferred manuscript. It will also be useful to readers of the reprint and help them to obtain a balanced view of the paper.

If you wish to submit a full revision, please use our "Full Revision" template. It is important to use the appropriate template to clearly inform the editors of your intentions.]

1. General Statements [optional]

This section is optional. Insert here any general statements you wish to make about the goal of the study or about the reviews.

We thank the reviewers for their positive appraisal of our manuscript and for their suggestions for improvement. The study highlights major roles for SRSF3 in coordinating expression and splicing of key cell cycle regulators, to underpin epicardial formation and EMT. Given the central importance of the epicardium in controlling cardiac development, and potential to facilitate regeneration, the epicardium has attracted much interest. We present the first endogenous iCLIP to map the SRSF3-RNA interaction network, which decisively confirms SRSF3 as a major cell cycle regulator. A vast number of target genes are presented, two of which are validated and characterized, *Ccnd1* and *Map4k4*, to exemplify transcriptional and splicing roles, respectively. An important finding by single-cell RNA-sequencing was the remarkable capacity for non-recombined epicardial cells to outcompete *Srsf3*-depleted epicardial cells and fully compensate for a substantial impairment in early development, highlighting the significant confounding effect of mosaic recombination on embryonic phenotyping. Previous studies using inducible Cre lines have failed to consider the impact of mosaic recombination, which has led to inaccurate phenotyping and conclusions. Beyond these major advances, we offer a comprehensive whole heart single cell RNA-seq data set at a key developmental stage for the heart (E13.5), complete with epicardial lineage tracing, which we anticipate will provide an invaluable resource to the research community. We are confident that our planned revision experiments will address all remaining substantive concerns raised by the reviewers, to elaborate on the mechanisms of SRSF3-controlled epicardial proliferation.

2. Description of the planned revisions

Insert here a point-by-point reply that explains what revisions, additional experimentations and analyses are planned to address the points raised by the referees.

Reviewer 1

Rev 1. Major comment 2 and 3. Stronger evidence of *Srsf3*/SRSF3 depletion.

As we no longer have access to *Srsf3* cKO mice, we cannot obtain fresh samples for western blotting, however we will perform RT-PCR to demonstrate the presence of recombined *Srsf3*, confirming exon 2/3 deletion, in the hearts of *Srsf3* cKO heterozygote embryos.

Rev 1. Major comment 4. Rescue of impaired proliferation by manipulating *Map4k4*.

Since overall expression of *Map4k4* was unaltered, we will investigate the effect of manipulating splicing at exons 16 and 17 on the proliferative capacity of *Srsf3* knockdown and control epicardial cells using splice modifying oligonucleotides. However, given that proliferation requires coordinated regulation at multiple levels, and SRSF3 is shown by irCLIP to bind a large number of key cell cycle regulators, we do not anticipate that restored splicing of a single gene will fully rescue the extreme reduction in proliferation of *Srsf3* knockdown cells.

Rev 1. Minor comment 1. Minor typographical and spacing errors.

We will correct and thoroughly check the revised manuscript.

Rev 1. Minor comment 2. Rationale for the use of two mouse models.

We will clarify the rationale, which relates to the timing, efficiency and specificity of *Srsf3* targeting. Briefly, the cKO enables efficient targeting of early progenitors within the proepicardium, but lacks specificity for later stage analysis, as Tg(Gata5-Cre) also targets a subset of cardiomyocytes, and is confounded by lethality from E12.5. The iKO enables more specific targeting of the epicardium and investigation of later developmental roles, although the restrictive window for induction by tamoxifen limits efficiency, giving rise to the mosaicism. Thus, while neither model is optimal, these are the best available epicardial Cre lines, and their combined use partly mitigates the inadequacies of the individual lines. We will ensure that these points are clear in the revised manuscript.

Rev 1. Minor comment 3. Stacked bar to accompany Figure 4A.

We will include the following stacked bar plot, which illustrates the reduction in mesenchymal epicardial derivatives, consistent with reduced EMT.

Rev 1. Minor comment 4. Quantification of *Srsf3*-expressing cells in Figure 4B.

Indeed, the retention of SRSF3 in the majority of tdTomato+ sorted cells seems counter-intuitive. Despite the widespread reliance upon Cre reporter lines, the highly efficient recombination of the short transcriptional stop sequence of Cre reporters does not accurately reflect recombination across multiple exons for gene deletion, which is typically much less efficient. As we show, *Srsf3*-depleted cells are compromised by impaired proliferation and viability. Some are lost *in vivo*, others likely do not adhere after sorting, and the enrichment of the *Srsf3*-retaining proliferative population would be further exaggerated after 48 hours in culture. We will include the following quantification alongside the images, which confirms an overall reduction in CyclinD1+ cells in *Srsf3* iKO and, importantly, that all SRSF3 negative cells lack CyclinD1.

It should be noted, however, that these images were taken as an additional control for scRNA-seq to evaluate extent of SRSF3 depletion, and represent a pooled sample from n=3 control hearts, n=2 *Srsf3* iKO hearts, assessed at a single time point after culture. More robust analyses of proliferation are presented in Figures 4 (*in vivo*) and 6 (*ex vivo*).

Reviewer 2

Rev 2. Major comment 1. Mechanism of hyper-proliferation in *Srsf3*-retaining cells

We suspect that all epicardial cells (not only those of mutant hearts) possess an intrinsic mechanism to sense epicardial cell density and adjust proliferation to reach a programmed setpoint and achieve complete coverage of the myocardium, via a mechanism that has yet to be determined. The suggestion of lateral inhibition by YAP/TAZ is a good one. Although this pathway appears to be minimally affected at the transcriptomic level (see below, only a modest decrease in epicardial *Tead2*), we will explore involvement by immunostaining for nuclear YAP, given the primary mode of regulation at the protein level. However, we recognise several other potential mechanisms e.g. paracrine/ECM signalling, that would be beyond the scope of the current study to investigate.

Rev 2. Major comment 2. Mechanism of elevated *Srsf3* expression in non-targeted mutant epicardium.

Given that *Srsf3* declines markedly in control embryos between E11.5 and E15.5, we will seek to clarify whether the elevated *Srsf3* in non-recombined mutant hearts actually represents an up-regulation of the gene or simply a retention compared with the control i.e. a failure to down-regulate. Alongside this, we will investigate the splicing of *Srsf3* and *Srsf4* in MEC1 cells, to determine any differential inclusion of the autoregulatory exon 4.

We can provide the following additional data to address the remaining points, specifically to demonstrate that *Srsf1/2* and *Ptbp1/2* expression levels are not significantly altered in the epicardial lineage (scRNA-seq from lineage traced E15.5 embryos).

Rev 2. Major comment 3. Full assessment of splicing changes in *Srsf3*-deficient cells.

We will perform bulk RNA-seq (exon-junction reads) of control and *Srsf3* knockdown MEC1 cells, for a comprehensive, unbiased analysis of SRSF3-dependent alternative splicing.

Rev 2. Major comment 4. Mechanism of *Srsf3*-controlled *Ccnd1* expression.

In control and *Srsf3* knockdown cells, we will assess *Ccnd1* RNA stability after transcriptional inhibition using actinomycin D, based on a protocol used in a previous SRSF3 study (Ratnadiwakara and Änkö, DOI:10.21769/BioProtoc.3072), to distinguish between altered transcription and mRNA stability of *Ccnd1*. The striking reduction in *Ccnd1* mRNA (Fig. 7H) already suggests that control is not predominantly at the level of translation or post-translation.

Rev 2. Major comment 5. Unbiased bioinformatic analysis of alternative splicing, alternative polyadenylation, and mRNA processing.

As in comment 3 above, we will perform an unbiased analysis following bulk RNA-seq, with exon-junction reads, of control and *Srsf3* knockdown MEC1 cells. Unfortunately, the equivalent analysis in primary cells is not feasible, due to the low efficiency of *Srsf3* recombination and elimination of *Srsf3*-depleted cells. An *in vitro* experiment permits the harvesting of *Srsf3* knockdown cells at a stage that captures alternative splicing, polyadenylation, processing and expression levels, prior to loss of viability. It is important to note that, although MEC1 cells were clonally selected for their continued replication, they have not been artificially immortalised. They were originally subcloned from primary cultures of E13.5 wild-type murine embryonic ventricles. A recent study demonstrated SRSF2-mediated alternative splicing changes in immortalised mutant cell lines were comparable to those in primary cells (Xu et al. (2022) *Blood Adv.* 12;6(7):2092-2106), thus we are confident of faithfully recapitulating the major alternative splicing and mRNA processing events.

Rev 2. Major comment 6. Contribution of *Map4k4* alternative splicing to the *Srsf3* knockdown phenotype.

Given that SRSF3 is shown by irCLIP to bind a large number of key cell cycle regulators, we agree that the differential splicing of *Map4k4* is unlikely to be sufficient to cause the extreme reduction in proliferation of *Srsf3* knockdown cells. Cell cycling requires coordinated regulation at multiple levels. *Map4k4* was presented as one example of differential splicing that is consistent with cell cycle exit in *Srsf3* knockdown cells, but is not unique in this regard. To investigate the contribution of *Map4k4* splicing to epicardial cell proliferation, and its control by SRSF3, we will attempt to modify exon 16/17 skipping/inclusion using splice-modifying oligonucleotides and assess the effect in proliferation assays.

Rev 2. Major comment 7. Evidence of senescence in *Srsf3* iKO embryos.

To support the observed changes in senescence-associated transcriptional changes, we will assess E13.5 embryonic sections for the presence of senescent cells by p16 and p21 immunostaining.

Rev 2. Major comment 8. Localisation and proliferation of WT1 positive /tdTom negative cells in the pro-epicardial organ.

We do not believe there is a regional enrichment of WT1 positive/tdTom negative epicardial cells within the PEO, rather since only non-targeted (tdTom negative) cells migrate onto the heart, they appear to be separating from the targeted (tdTom positive) cells. As we described, PEO explant culture permits more accurate quantification of cells, compared with assessment of a 3D structure across multiple individual sections. Therefore, we will re-assess Ki67 staining in PEO explants, using a purpose built CellProfiler pipeline, and express as %Ki67+/tdTom+ versus %Ki67+/tdTom- cells.

Rev 2. Major comment 10. Quantification of *Srsf3*-expressing Cyclin D1+/- cells in Figure 4B.

We will include the following quantification of the images, which confirms an overall reduction in

CyclinD1+ cells in *Srsf3* iKO and, importantly, that all SRSF3 negative cells lack CyclinD1.

It should be noted, however, that these images were taken as an additional control for scRNA-seq to evaluate extent of SRSF3 depletion, and represent a pooled sample from n=3 control hearts, n=2 *Srsf3* iKO hearts, assessed at a single time point after culture. More robust analyses of proliferation are presented in Figures 4 (*in vivo*) and 6 (*ex vivo*).

Rev 2. Minor comments 1-5.

We will fully address these comments by correcting typographical errors and clarifying the targeted cell type for comment 5.

Reviewer 3

Rev 3. Major comment 1. Clarification of cardiomyocyte targeting in Tg(Gata5-Cre), and expanding previously reported roles.

We will highlight the reported role for SRSF3 in capping of cardiomyocyte transcripts, in addition to roles in proliferation, in the revised manuscript. With regard to the cardiomyocyte subsets targeted by the Tg(Gata5-Cre), we will not duplicate the previously published quantification, which reported Cre-based reporter labelling to be >50% of both the right and left ventricular wall (Sedmera et al., 2024; PMID: 39435333). Their observations of “patchy” cardiomyocyte targeting aligns very well with our findings and those presented previously ((PMID: 39435333; PMID: 32068311)). We shall instead cite these studies and highlight the patchy targeting.

Rev 3. Major comment 2. Expanding on the nature of the myocardial compaction defect.

We can include the data below from the scRNA-seq studies, along with any insights that emerge from the MEC1 RNA-seq e.g. relating to disrupted epicardial paracrine signals that have been implicated in cardiomyocyte proliferation e.g. *Rspo1*, Wnt, retinoic acid, which are also identified as SRSF3 targets by irCLIP.

Based on markers used by Rhee et al. (2021; PMID: 34279605) and others, the up-regulation of *Nppa* and *Nppb*, along with down-regulation of *Ttn* and *Fabp3*, in all cardiomyocyte subsets, and decreased *Top2a* and *Pcna* specifically in the proliferating cardiomyocytes, are consistent with the non-compaction phenotype in *Srsf3* iKO hearts.

Rev 3. Major comment 4. *Srsf3* regulation by itself and hypoxia.

We will include irCLIP data (as below) to demonstrate that SRSF3 binds *Srsf3* transcripts in the epicardium, consistent with previous reports, thus may be responsible for modulating its own expression levels.

Given that *Srsf3* declines markedly in control embryos between E11.5 and E15.5, we will seek to clarify whether the elevated *Srsf3* in non-recombined mutant hearts actually represents an up-regulation of the gene or simply a retention compared with the control i.e. a failure to down-regulate. Alongside this, we will investigate the splicing of *Srsf3* and *Srsf4* in MEC1 cells, to determine any differential inclusion of the autoregulatory exon 4. We will also assess hypoxic embryonic hearts, to determine whether or not hypoxia directly influences SRSF3 levels.

Rev 3. Major comment 5. Mechanism of *Srsf3*-controlled *Ccnd1* expression.

We can provide data to confirm that Cyclin D1 is indeed significantly reduced in *Srsf3* knockdown MEC1 cells. Moreover, in these cells, we will assess *Ccnd1* RNA stability after transcriptional inhibition using actinomycin D, based on a protocol used in a previous SRSF3 study (Ratnadiwakara and Änkö, DOI:10.21769/BioProtoc.3072), to distinguish between altered transcription and mRNA degradation.

Rev 3. Comment 6. Brightening of immunofluorescence images

We can brighten the immunofluorescence panels shown in Fig. 1A, for example as below, although this naturally also increases background fluorescence level. Given that higher

magnification insets are included, and the resolution allows for considerable magnification to clearly discern nuclear signal, our preference would be not to increase the brightness excessively.

Rev 3. Comment 7. Removal of the word 'key' from the title

We would agree to removing the word 'key' if necessary, even though its use is entirely consistent with the phenotype and the enrichment of cell cycle-related genes by irCLIP (and, we anticipate further confirmation by the planned RNA-seq). Being a key regulator of proliferation does not preclude additional roles in other processes. How it ranks in importance relative to other regulators of epicardial cell cycle is an open question, but the striking correlations between *Srsf3* and cycling (in Figs 2G-H, 3D-G, 4B, D-F, 6D-E, 7I, 9D) suggest that it is indispensable.

3. Description of the revisions that have already been incorporated in the transferred manuscript

Please insert a point-by-point reply describing the revisions that were already carried out and included in the transferred manuscript. If no revisions have been carried out yet, please leave this section empty.

We present the data that are already available within the review plan, rather than revise the manuscript at this stage, given that the revisions we propose will require the addition of multiple new data panels into the figures. We will be better placed to design appropriately revised figures once all the additional data are available.

4. Description of analyses that authors prefer not to carry out

Please include a point-by-point response explaining why some of the requested data or additional analyses might not be necessary or cannot be provided within the scope of a revision. This can be due to time or resource limitations or in case of disagreement about the necessity of such additional data given the scope of the study. Please leave empty if not applicable.

Rev 1. Major comment 1. *Wt1* mRNA expression over time

We did not present *Wt1* mRNA expression in Fig. 1 A for comparison with *Srsf3*. Although epicardial *Wt1* declines over the course of development, coronary endothelial cells begin to express *Wt1* from E11.5, increasing thereafter (examples in panel C below). Thus, whole heart qRT-PCR would not show a decline comparable to *Srsf3*. However, when we previously visualised epicardial *Wt1* by single-molecule RNA in situ hybridization (RNAscope) and quantified expression using a flow cytometry-based RNA ISH method (PrimeFlow RNA Assay), a highly comparable decline in expression was observed. The data below are taken from our publication Lupu et al (2020) *Stem Cell Reports*. 14(5):770-787. We will refer to these data in the revised manuscript.

NOTE: Figure provided for reviewer has been removed. It showed selected panels from Figure 4 from Lupu, I.E., Redpath, A.N., Smart, N. (2020) Spatiotemporal Analysis Reveals Overlap of Key Proepicardial Markers in the Developing Murine Heart. *Stem Cell Reports*. 14, 770-787. doi: 10.1016/j.stemcr.2020.04.002. Epub 2020 Apr 30. PMID: 32359445; PMCID: PMC7221110.

Rev 2. Major comment 9. Assessing potential migration defect in *Srsf3* depleted epicardial cells.

The heading on page 6 refers to failure of progenitor cells to translocate from the PEO onto the heart (as evidenced in Figs 1F,G and 2D). The current consensus is that PEO translocation onto the heart is not actually a process of migration, but instead reflects the differential cellular outgrowth, requiring a high rate of proliferation (Rodgers et al., 2007; Kuhn and Liebherr, 1988; Wessels and Perez-Pomares, 2004; Hirose et al., 2006). The phenotype of failed translocation in *Srsf3* cKO is entirely consistent with the prominent role of SRSF3 in cell proliferation. Our data e.g. irCLIP and scRNA-seq do not support a major involvement in cell migration. To avoid confusion, we will use the term translocation of the PEO onto the heart, instead of migration, in the revised manuscript.

In vitro migration assays in *Srsf3* knockdown cells would be confounded by reduced viability impairing migration independently of any SRSF3-controlled mechanisms. Unless the new MEC1 RNA-seq data strongly supports a role for SRSF3 in controlling cell migration, we do not believe this experiment would be rational or informative.

Rev 3. Major comment 3. Cell competition by non-recombined cells.

The phenomenon we describe (elimination of cells that have undergone cell cycle arrest, apoptosis and senescence, and their complete replacement by the remaining non-targeted cells that continue to proliferate) is not consistent with the classical definition of cell competition, as mediated by *Myc*, which is rather “a tissue homeostasis mechanism by which low-anabolizing - but otherwise viable- cells are eliminated from tissues due to confrontation with higher-anabolizing cells”. Consistent with this, we do not find evidence of *Myc* up-regulation in the epicardial lineage, as shown below by scRNA-seq. Thus, an investigation of cell competition is neither logical nor supported by our data. We will edit the manuscript to use the terms eliminate/replace, rather than out-compete.

Original submission

First decision letter

MS ID#: dev.204918

MS Title: SRSF3 is a key regulator of epicardial formation

Authors: Irina-Elena Lupu; Susann Bruche; Anob M. Chakrabarti; Ian R. McCracken; Tamara Carsana; Andia Redpath; Nicola Smart

Dear Dr Smart,

Thank you for sending your manuscript to Development through Review Commons.

I have now received all the referees' reports on the above manuscript, and have reached a decision. The referees' comments are appended below, or you can access them online: please go to:

As you will see, the referees express considerable interest in your work, but have some significant criticisms and recommend a substantial revision of your manuscript before we can consider publication. If you are able to revise the manuscript along the lines suggested, which may involve further experiments, I will be happy receive a revised version of the manuscript. Your revised paper will be re-reviewed by one or more of the original referees, and acceptance of your manuscript will depend on your addressing satisfactorily the reviewers' major concerns. Please also note that Development will normally permit only one round of major revision. If it would be helpful, you are welcome to contact us to discuss your revision in greater detail. Please send us a point-by-point response indicating your plans for addressing the referees' comments, and we will look over this and provide further guidance.

Please attend to all of the reviewers' comments and ensure that you clearly highlight all changes made in the revised manuscript. Please avoid using 'Tracked changes' in Word files as these are lost in PDF conversion. I should be grateful if you would also provide a point-by-point response detailing how you have dealt with the points raised by the reviewers in the 'Response to Reviewers' box. If you do not agree with any of their criticisms or suggestions please explain clearly why this is so.

Reviewer 1

Summary:

The study by Irina Lupu and colleagues highlights SRSF3 as a key regulator of epicardial development by regulating epicardial cell proliferation. This was demonstrated via two murine knockout models; the first to assimilate the role SRSF3 plays in epicardial formation as a whole, and the second to address its importance post a pivotal maturation point. Through scRNA sequencing and irCLIP, several SRSF3 targets were ascertained and identified as cell cycle regulators. Those epicardial cells that did not lose SRSF3 compensated the loss of some of their mates by increasing SRSF3 expression and over-proliferating. Overall, the paper is interesting and the conclusions are largely supported by the provided data.

Major comments:

1. Authors claim that a "reduction in SRSF3 expression levels coincided with the downregulation of WT1 in the epicardium". This was evidenced by immunofluorescence imaging (figure 1A). I suggest conducting a qRT-PCR to quantify Wt1 expression over time, similar to the experiment they performed in figure 1B.
2. A western blot depicting SRSF3 protein production in controls compared to the knockout model may provide stronger evidence of its depletion (figure 1E).
3. Authors state that they were unable to directly identify the absence of exons 2 and 3 in individual cells. Please provide evidence that exons 2 and 3 have been knocked out, at least by performing a qRT-PCR.
4. To prove the functional implication in the observed phenotype of the identified SRSF3 targets, please interfere with Map4k4 activity or expression and check whether the defective epicardial cell proliferation is reverted. This should be done at least in vitro, ideally in vivo.

Minor comments:

1. Several minor typos and spacing issues were observed. Please correct.
2. It would be good for the reader if the authors would simplify their rationale for the use of the two mouse models. It is slightly convoluted and not easy to follow.
3. In figure 4, it is recommended to add a stacked bar plot to represent the percentage of each cell cluster/population after 4A. This would help the reader
4. Figure 4B. It is confusing for the reader to understand the fact that the majority of tdTomato+ sorted cells in Srsf3 iKO keep expressing Srsf3. Including the quantification of the image could help.

Reviewer 2

In this study Lupu and colleagues convincingly demonstrate that SRSF3 as an important regulator of cell cycle in the epicardium. This is shown by two series of conditional deletion experiments and molecular analyses and the work is well documented and clearly presented. The proliferation role appears to be consistent with the role of SRSF3 in other cell types. In addition the authors observed epicardial recovery by unrecombined cells and make an important general point about mosaicism and phenotyping in conditional mutagenesis experiments. The authors have discussed in detail how they plan to address an earlier set of comments. Their replies are well argued and fair, and the proposed revisions include useful experiments such as identifying splicing changes in mutant hearts using bulk RNA sequencing and investigating the origin of *Srsf3* expressing cells in mosaic hearts. I have only a small number of additional points.

1. Can the authors rule out an indirect role in myocardial proliferation that might contribute to the compaction defect using the later activated Cre line? The authors could expand the last paragraph of the discussion to compare the new data with the known role of SRSF3 in regulating proliferation in other contexts.

2. Concerning *Myc* and cell competition, the authors demonstrate that *Myc* transcripts are bound by SRSF3 (page 11), suggesting that transcript processing may be impacted. They should discuss the possibility that this might place cells with normal *Myc* processing at an advantage.

3. The title as it stands, with or without the word "key", is not very informative. Perhaps the authors can extend it, for example mentioning epicardial proliferation.

First revision

Author response to reviewers' comments

We thank the reviewers for their positive appraisal of our manuscript and for their suggestions for improvement.

The study highlights major roles for SRSF3 in coordinating expression and splicing of key cell cycle regulators, to underpin epicardial formation and EMT. Given the central importance of the epicardium in controlling cardiac development, and potential to facilitate regeneration, the epicardium has attracted much interest. We present the first endogenous iCLIP to map the SRSF3-RNA interaction network, which decisively confirms SRSF3 as a major cell cycle regulator. A vast number of target genes are presented, two of which are validated and characterized, *Ccnd1* and *Map4k4*, to exemplify transcriptional and splicing roles, respectively.

An important finding by single-cell RNA-sequencing was the remarkable capacity for non-recombined epicardial cells to outcompete *Srsf3*-depleted epicardial cells and fully compensate for a substantial impairment in early development, highlighting the significant confounding effect of mosaic recombination on embryonic phenotyping. Previous studies using inducible Cre lines have failed to consider the impact of mosaic recombination, which has led to inaccurate phenotyping and conclusions. Beyond these major advances, we offer a comprehensive whole heart single cell RNA-seq data set at a key developmental stage for the heart (E13.5), complete with epicardial lineage tracing, which we anticipate will provide an invaluable resource to the research community.

We have now addressed all substantive concerns raised by the reviewers. In summary, this includes:

- i) elaborating the mechanisms of SRSF3-controlled epicardial proliferation by conducting an unbiased analysis of SRSF3-mediated transcription, splicing and RNA polyadenylation.
- ii) providing definitive evidence of *Srsf3* exon 2/3 deletion after recombination by *Wt1CreERT2*.
- iii) distinguishing a role for SRSF3 in controlling transcription, rather than stability, of *Ccnd1*.

- iv) elaborating on the mechanism for compensatory hyperproliferation.
- v) clarifying the rationale for the epicardial Cre lines used in our study.
- vi) providing further validation and quantification where requested.
- vii) correcting typographical errors.

This adds 1 main figure, 6 Supplementary figures and 15 Tables, in addition to numerous new data panels to existing figures.

Please find below a point-by-point response to the comments of the original *Review Commons* reviewers and those of the *Development* reviewers.

Review Commons Reviewers comments:

Reviewer #1 (Evidence, reproducibility and clarity (Required)):

Summary: The study by Irina Lupu and colleagues highlights SRSF3 as a key regulator of epicardial development by regulating epicardial cell proliferation. This was demonstrated via two murine knockout models; the first to assimilate the role SRSF3 plays in epicardial formation as a whole, and the second to address its importance post a pivotal maturation point. Through scRNA sequencing and irCLIP, several SRSF3 targets were ascertained and identified as cell cycle regulators. Those epicardial cells that did not lose SRSF3 compensated the loss of some of their mates by increasing SRSF3 expression and over-proliferating. Overall, the paper is interesting and the conclusions are largely supported by the provided data.

Major comments:

1. Authors claim that a "reduction in SRSF3 expression levels coincided with the downregulation of *Wt1* in the epicardium". This was evidenced by immunofluorescence imaging (figure 1A). I suggest conducting a qRT-PCR to quantify *Wt1* expression over time, similar to the experiment they performed in figure 1B.

We did not present *Wt1* mRNA expression in Fig. 1 A for comparison with *Srsf3* for two reasons:

1. Although epicardial *Wt1* declines over the course of development, coronary endothelial cells begin to express *Wt1* from E11.5, increasing thereafter (examples in panel C below). Thus, whole heart qRT-PCR would not show a decline comparable to *Srsf3*.
2. The downregulation of epicardial *Wt1* over the course of development has already been extensively described in our 2020 manuscript (Lupu et al (2020) *Stem Cell Reports*. 14(5):770-787), in which we visualised epicardial *Wt1* by single-molecule RNA in situ hybridization (RNAscope) and quantified expression using a flow cytometry-based RNA ISH method (PrimeFlow RNA Assay). We now explicitly highlight the comparable decline in *Wt1* and cite this study on page 5 of the revised manuscript. For the reviewer's reference, the relevant figure panels are shown here:

NOTE: Figure provided for reviewer has been removed. It showed selected panels from Figure 4 from Lupu, I.E., Redpath, A.N., Smart, N. (2020) Spatiotemporal Analysis Reveals Overlap of Key Proepicardial Markers in the Developing Murine Heart. *Stem Cell Reports*. 14, 770-787. doi: 10.1016/j.stemcr.2020.04.002. Epub 2020 Apr 30. PMID: 32359445; PMCID: PMC7221110.

2. A western blot depicting SRSF3 protein production in controls compared to the knockout model may provide stronger evidence of its depletion (figure 1E).

and

3. Authors state that they were unable to directly identify the absence of exons 2 and 3 in individual cells. Please provide evidence that exons 2 and 3 have been knocked out, at least by performing a qRT-PCR.

Whole heart western blot to demonstrate epicardial SRSF3 depletion would be diluted by expression in the more numerous cardiomyocytes. Moreover, we no longer have access to *Srsf3* cKO mice to obtain fresh samples for western blotting.

To address this important point, we reanalysed the *Srsf3* iKO single cell RNA-seq data using a customised script that enabled us to demonstrate the recombination and deletion of exon 2/3 in mutant epicardial cells. These data are included in a new Supplementary Figure 4. We maintain that the 3' bias of 10X chromium technology limits the ability to detect recombination in all transcripts, and that this approach is further confounded by the failure of targeted cells to proliferate and to pass standard quality control filters. Thus, the figure of ~10% recombination in mutant epicardial cells is likely severely under-estimated, but importantly validates epicardial deletion with our inducible genetic knockout strategy.

4. To prove the functional implication in the observed phenotype of the identified SRSF3 targets, please interfere with *Map4k4* activity or expression and check whether the defective epicardial cell proliferation is reverted. This should be done at least in vitro, ideally in vivo.

We appreciate the reviewer's comment and the importance of functional validation for implicating SRSF3 targets in epicardial function. However, since overall expression of *Map4k4* was unaltered, epicardial proliferation could not be reverted simply to restoring its expression. Moreover, given that proliferation requires coordinated regulation at multiple levels, and SRSF3 is shown by irCLIP to bind hundreds of key cell cycle regulators, we do not anticipate that restored expression/splicing of a single gene will sufficiently rescue the extreme reduction in proliferation of *Srsf3* knockdown cells. We have made clear that we do not feel this mechanism fully accounts for proliferative arrest in the concluding sentence of that section: "*suggesting that regulation [of Map4k4 splicing] by SRSF3 may, in part, contribute to the switch of epicardial cells from the proliferative to senescent state.*"

Minor comments:

1. Several minor typos and spacing issues were observed. Please correct.

We have carefully reviewed and corrected these errors. Thank you for highlighting them.

2. It would be good for the reader if the authors would simplify their rationale for the use of the two mouse models. It is slightly convoluted and not easy to follow.

We have included a simple explanation and comparison between the two mouse models in the revised manuscript which explicitly states the advantages and limitations of each.

On page 6...*Despite the ST and cardiomyocyte recombination, Tg(Gata5-Cre) enables efficient targeting of E9.5 proepicardial progenitors, that cannot be achieved with inducible epicardial Cre lines.*

On page 8 (in relation to the Wt1CreERT2 line)...*The restricted window for induction prioritises specificity over efficiency. While cEC and cardiomyocyte targeting can be largely avoided with this strategy, it produces mosaic recombination, even with our optimised tamoxifen regime. Thus, while both constitutive and inducible epicardial Cre lines are imperfect, they are considered the best available, and their combined use partly mitigates their respective inadequacies.*

3. In figure 4, it is recommended to add a stacked bar plot to represent the percentage of each cell cluster/population after 4A. This would help the reader.

We now include a stacked bar plot (Figure 4B) to represent the percentage of each cell cluster.

4. Figure 4B. It is confusing for the reader to understand the fact that the majority of tdTomato+ sorted cells in *Srsf3* iKO keep expressing *Srsf3*. Including the quantification of the image could help.

Indeed, the retention of SRSF3 in the majority of tdTomato+ sorted cells seems counter-intuitive. Despite the widespread reliance upon Cre reporter lines, the highly efficient recombination of the short transcriptional stop sequence of Cre reporters does not accurately reflect recombination across multiple exons for gene deletion, which is typically much less efficient, as widely reported (Fernandez-Chacon et al., 2019; Payne et al., 2018; Redpath and Smart, 2021). Moreover, as we show, *Srsf3*-depleted cells are compromised by impaired proliferation and viability. Some are lost in vivo, others likely struggle to adhere after sorting, and the enrichment of the *Srsf3*-retaining proliferative population would be further exaggerated after 48 hours in culture. These limitations thus result in an underestimation of *Srsf3*-depleted cells.

We now include quantification of these data alongside the images (Figure 4D), which confirms an overall reduction in CyclinD1+ cells in *Srsf3* iKO and, importantly, that all SRSF3 negative cells lack CyclinD1.

Reviewer #1 (Significance (Required)):

The paper will be of interest to readers in the field of cardiology, embryology and molecular biology. It will advance the field especially in the study of the development of the epicardium. The models are sophisticated and the experiments carefully performed. My field is molecular cardiology, with interest in RNA-binding proteins.

Reviewer #2 (Evidence, reproducibility and clarity (Required)):

In this study, Lupu et al. analyzed the role of the RNA-binding protein SRSF3 for epicardial development. The authors found that *Srsf3* is highly expressed in the proepicardial organ and during early stages of epicardial layer formation. Conditional inactivation of SRSF3 in the proepicardial organ stage using a *Gata5*-Cre driver line resulted in defective formation of the epicardium, accompanied by a proliferation arrest of the proepicardium, resulting in embryonic lethality at E12.5. In contrast, epicardial-specific *Srsf3* deletion at later stages using the inducible *Wt1*CreERT2 line caused a less severe phenotype indicated by impaired coronary vasculature formation, reduced cardiac compaction, and myocardial hypoxia. Mosaic recombination yielded a small population of epicardial cells that upregulate *Srsf3*, hyperproliferate and compensate for the depleted *Srsf3* negative lineage. Single-cell RNA sequencing of control and epicardial *Srsf3* knock out hearts, combined with infrared CLIP to map SRSF3 binding sites in the transcriptome identified a number of putative SRSF3 targets involved in mitotic cell cycle control. Among others, SRSF3 binds directly to transcripts encoding key regulators of proliferation, such as Cyclin D1, and senescence, including MAP4K4. The authors conclude that SRSF3 exerts different functions in processing of RNAs, including splicing.

Overall, this is a well-written and well-organized manuscript, describing interesting findings in the field of epicardial development. However, the mechanistic part is not overly strong. The authors detected some moderate changes in the distribution of different *Map4k4* splicing isoforms after knockdown of *Srsf3* in an immortalized epicardial cell line but did not go any deeper. The cause for the reduced presence of transcripts for the SRSF3-target *Ccnd1* after knockdown of *Srsf3* remains enigmatic.

Reviewer #2 (Significance (Required)):

The authors raise a number of speculations why remaining *Srsf3*-expressing cell start to hyperproliferate after inactivation of *Srsf3* but it does not become clear which mechanism is critical. How do non-targeted epicardial cells in the mosaic recombination model sense the loss of SRSF3 knock out cells, resulting in hyperproliferation and enhanced *Srsf3* expression? Is a loss of lateral inhibition, e.g. by activated YAP/TAZ, causative for enhanced proliferation of the remaining epicardial cells and an elevated expression level of WT1 and SRSF3? Immunofluorescence staining and/or qRT-PCR of YAP/TAZ and TEADs might provide an answer.

We suspect that all epicardial cells (not only those of mutant hearts) possess an intrinsic mechanism to sense epicardial cell density and adjust proliferation to reach a programmed setpoint and complete coverage of the myocardium, via a mechanism that has yet to be determined. The suggestion to investigate lateral inhibition by YAP/TAZ is eminently sensible. However, we found minimal support for this mechanism at the transcriptional level (only a modest decrease in epicardial *Tead2* in *Srsf3* iKO cells by scRNAseq and no change in *Tead1,3,4*, *Yap1* or *Taz*). Nor did we detect any alteration in active (non-LATS1/2 phosphorylated), nuclear-localised YAP1 protein. These new data can be found in Supplementary Fig. 12, along with data which exclude a role for a MYC-mediated cell competition mechanism.

As outlined below, we now offer evidence to support the auto-regulation of *Srsf3* splicing to control its own expression levels as the primary mechanism driving compensatory hyper-proliferation.

Is the elevated expression of *Srsf3* in non-targeted epicardial cells due to enhanced transcription and/or by altered post-transcriptional processes? How does this observation fit to previous reports indicating that *Srsf3* overexpression promotes inclusion of an autoregulatory cassette exon (exon 4) containing a premature (in-frame) stop codon in *Srsf3*, thereby confining this SRSF3 isoform to nonsense mediated decay (NMD) (doi: 10.1093/emboj/16.16.5077, doi: 10.1186/gb-2012-13-3-r17, doi: 10.1038/srep14548, doi: 10.1161/CIRCRESAHA.118.31451)? In contrast, *Srsf1* as well as *PTBP1/2* have been previously reported to regulate *Srsf3* expression by promoting exon 4 skipping. The authors should perform RNA seq and/or qRT-PCRs validation to check the inclusion of *Srsf4* Exon4 as well as *Srsf1* and *PTBP1/2* expression levels in control and knock out epicardial cells.

To address this concern, we investigated *Srsf3* splicing in MEC1 epicardial cells. While control-transfected MEC1 included auto-regulatory exon 4 in 10.66% of *Srsf3* transcripts, exon 4 was present in 1.67% (si*Srsf3*#1) and 3.09% (si*Srsf3*#2) of transcripts from *Srsf3* knockdown MEC1 cells (Supplementary Fig. 13A). Consistent with the notion of auto-regulation, we detected extensive SRSF3 CLIP-tag clusters across the autoregulatory cassette exon 4 of *Srsf3* (Supplementary Fig. 13B), supporting a mechanism for *Srsf3* auto-up-regulation in surviving cells following *Srsf3* iKO or knockdown. In contrast, the autoregulatory exons of *Srsf4*, which are known not to be cross-regulated by SRSF3 (Ånkö et al., 2012), were not differentially spliced in *Srsf3* knockdown MEC1 cells (Supplementary Fig. 13A). We also confirmed that *Srsf1*, *Ptbp1* and *Ptbp2* transcription was not altered in *Srsf3* iKO epicardial cells (Supplementary Fig. 13C), collectively supporting auto-regulation by SRSF3 itself as the predominant mechanism driving elevated expression and hyperproliferation.

It remains unclear by which mechanisms (alternative splicing, alternative polyadenylation, miRNA processing, or others) SRSF3 mainly exerts its function in the embryonic epicardial lineage. The selection and validation of *Map4k4* as a splicing target is not based on an unbiased splicing analysis. In my opinion it is mandatory to provide a full assessment of splicing changes in *Srsf3*-deficient cells, either by long-range sequencing or by analysis of exon-junction reads.

We conducted an unbiased analysis of transcription, splicing and polyadenylation in MEC1 epicardial cells, demonstrating significant SRSF3-dependent roles in each process through *Srsf3* knockdown. These data are presented in Figure 7, Supplementary Figure 7 and Tables 2-16. The main conclusions, namely that SRSF3 controls splicing (prominently via alternative exon inclusion) and cell cycle in the epicardium are described in full in the manuscript on page 12.

Likewise, it is completely enigmatic what SRSF3 does to *Ccnd1* transcripts. Does SRSF3 increase the half-life of *Ccnd1*, does it impact trafficking? At least, the authors have to determine changes in the half-life of *Ccnd1* after depletion of SRSF3.

We conducted mRNA stability assays to measure *Ccnd1* decay, alongside that of the G1/S regulator *Ccng1*, after inhibiting transcription with Actinomycin D. We found no significant differences in half-life or decay rate constant for either gene after knockdown of *Srsf3* (Supplementary Fig. 9), suggesting that SRSF3 controls cyclin D1 transcription rather than its stability.

An unbiased bioinformatics analysis addressing alternative splicing, alternative polyadenylation, and mRNA processing is necessary. Ideally, primary epicardial cells should be used and not an

immortalized epicardial cell lines. It is well known, that splicing in cell lines differs substantially from splicing in primary cells.

As described above, we conducted an unbiased analysis of transcription, splicing and polyadenylation in MEC1 epicardial cells, demonstrating significant SRSF3-dependent roles in each process through *Srsf3* knockdown. Unfortunately, the equivalent analysis in primary cells is not feasible, due to the low efficiency of *Srsf3* recombination and elimination of *Srsf3*-depleted cells. An *in vitro* experiment permits the harvesting of *Srsf3* knockdown cells at a stage that captures alternative splicing, polyadenylation, processing and expression levels, prior to loss of viability. It is important to note that, although MEC1 cells were clonally selected for their continued replication, they have not been artificially immortalised. They were originally subcloned from primary cultures of E13.5 wild-type murine embryonic ventricles. A recent study demonstrated SRSF2-mediated alternative splicing changes in immortalised mutant cell lines were comparable to those in primary cells (Xu et al. (2022) *Blood Adv.* 12;6(7):2092-2106). This and the independent corroboration of the same key changes that we described *in vivo* provides confidence that we can faithfully recapitulate the major alternative splicing and mRNA processing events in MEC1 cells.

I am not convinced that the moderate changes of different *Map4k4* splicing isoforms after knockdown of *Srsf3* are really responsible for the rather drastic phenotype. Additional experiments are needed to prove a decisive function of a shift in *Map4k4* splicing isoforms for hyperproliferation of epicardial cells.

We appreciate the reviewer's comment and the importance of functional validation for implicating SRSF3 targets in epicardial function. Given that proliferation requires coordinated regulation at multiple levels, and SRSF3 is shown by irCLIP to bind hundreds of key cell cycle regulators, we agree that restored splicing of a single gene will likely be insufficient to rescue the extreme reduction in proliferation of *Srsf3* knockdown cells. We have made clear that we do not feel this mechanism fully accounts for proliferative arrest in the concluding sentence of that section: "*suggesting that regulation [of *Map4k4* splicing] by SRSF3 may, in part, contribute to the switch of epicardial cells from the proliferative to senescent state.*"

The authors claim that inactivation of *Srsf3* inhibits cell proliferation and causes a senescence-like phenotype. The claim for acquisition of senescence is solely based on transcriptional changes. No attempts were made to visualize an increase of senescent cells in *Srsf3*-mutant embryos. The authors need to perform SA-bGAL assays or use other techniques to analyse the appearance of senescent cells in the mutants.

To support the observed changes in senescence-associated transcriptional changes, we now include data from E13.5 embryonic sections demonstrating an increased presence of p21-positive senescent epicardial cells by immunostaining (Figure. 4I-J).

Fig. 2E indicates that WT1 positive / tdTom negative epicardial cell population is enriched in a specific region of the pre-epicardial organ from *Srsf3*KOs. However, it is not clear whether these cells proliferate. The authors should quantify Ki67positive cells in both the WT1-positive / tdTom-positive and the WT1positive / tdTom-negative epicardial population.

We do not believe there is a regional enrichment of WT1-positive/tdTom-negative epicardial cells within the PEO, rather since only non-targeted (tdTom-negative) cells migrate onto the heart, they appear to be separating from the targeted (tdTom-positive) cells. PEO explant culture permits more accurate quantification of cells, compared with assessment of a 3D structure across multiple individual sections. We believe our assessment of Ki67+ proliferative PEO cells provides a more robust set of data to address this concern (Figure 2).

In the headline on page 6, the authors stated that "SRSF3 depletion in the PEO results in impaired ... migration of epicardial progenitor cells", which they deduced from the reduced outgrowth of ventricular epicardial explants. However, the reduced outgrowth from the PEO could be caused by both, reduced proliferation and/or reduced migration. Therefore, the authors should provide additional data clearly indicating reduced migration, e.g. by blocking transcription. Scratch assays of SRSF3 knockout/knockdown vs. control epicardial cells would strengthen the analysis, Is there a change in the GO term "regulation of migration"?

The heading on page 6 refers to failure of progenitor cells to translocate from the PEO onto the heart (as evidenced in Figs 1F,G and 2D). The current consensus is that PEO translocation onto the heart is not actually a process of migration, but instead reflects differential cellular outgrowth, driven in part by a high rate of proliferation (Rodgers et al., 2007; Kuhn and Liebherr, 1988; Wessels and Perez-Pomares, 2004; Hirose et al., 2006). The phenotype of failed translocation in *Srsf3* cKO is entirely consistent with the prominent role of SRSF3 in cell proliferation. Our irCLIP, RNA-seq and scRNA-seq data do not support a major involvement in cell migration. To avoid confusion, we have used the term translocation of the PEO onto the heart, instead of migration, in the revised manuscript.

To prove the reduced proliferation ratio in Figure 4B, quantification of Cyclin D1 positive cells in both SRSF3 positive and negative cells is required.

We now include quantification of these data alongside the images (Figure 4D), which confirms an overall reduction in CyclinD1+ cells in *Srsf3* iKO and, importantly, that all SRSF3-negative cells lack CyclinD1.

Minor issues

Abstract line 12: a full stop is missing at the end of the sentence.

Figure 1A: E11.5, figure label 'DAPI WT1' is missing.

Page 8: Bracket in front of Fig. 4B is missing

Page 8: G2M phase change uniformly to G2/M phase

Page 9: '*Srsf3*-depleted hearts also demonstrated an increased abundance of epicardial cells with upregulated expression of genes associated with quiescence, such as *Clu48*, and senescence, for example *Map4k4*, *Tmem30a* and *Pofut249* (Fig. 4F)'. The sentence is misleading, implying *Srsf3* inactivation in all cardiac cell types ('*Srsf3*-depleted hearts').

We thank the reviewer for careful reviewing our manuscript and for alerting us to these errors. They have now all been corrected.

Reviewer #3 (Evidence, reproducibility and clarity (Required)):

Lupu et al have identified a role for the RNA binding protein SRSF3 in epicardial development. In a well written manuscript containing many experiments the authors show that this protein is required at different time points in epicardial development to control a range of processes, in particular cell proliferation. This advances understanding of the complex roles of this RNA binding protein in the heart - and raises an important message about how incomplete Cre recombination needs to be considered in interpreting conditional mutant phenotypes. The following points should be addressed.

1. SRSF3 is known to play essential developmental roles in the myocardium where it regulates capping of transcripts involved in contraction. This point should be mentioned in addition to roles in proliferation. To facilitate understanding, the authors should say more about the subset of cardiomyocytes labelled by *Gata5-Cre*. For example, is this the result of stochastic activation of the transgene or is a specific subset of cells labelled? How much of the myocardium is targeted?

We fully agree with the first point, and have included the following sentence in our Introduction: *SRSF3 was shown to be required for cardiomyocyte proliferation during development, and to critically prevent de-capping of contraction-related mRNA transcripts, with conditional deletion of SRSF3 from this lineage resulting in embryonic lethality mid-gestation*(Ortiz-Sánchez et al., 2019).

With regard to the cardiomyocyte subsets targeted by the Tg(*Gata5-Cre*), we cannot duplicate the previously published quantification, which reported Cre-based reporter labelling to be >50% of both

the right and left ventricular wall (Sedmera et al., 2024; PMID: 39435333). Their observations of “patchy” cardiomyocyte targeting aligns very well with our findings and those presented previously ((PMID: 39435333; PMID: 32068311)). We have instead cited these studies and highlighted the patchy targeting on page 6 of the revised manuscript.

2. The authors show failure of ventricular compaction at E13.5 using *Wt1-CreERT2* and go on to assess proliferation in epicardial cells. As epicardial-derived signals are known to promote compact myocardial growth, they should also show whether there are indirect defects in proliferation in compact layer myocardium that might explain the non-compact phenotype. The authors should also indicate if any of the large number of genes bound by SRSF3 encode known or potential pro-proliferative signals from the epicardium or EPDCs to the myocardium and potentially validate their altered expression in mutant hearts.

Based on markers used by Rhee et al. (2021; PMID: 34279605) and others, we now include data to demonstrate the up-regulation of *Nppa* and *Nppb*, along with down-regulation of *Ttn* and *Fabp3*, in all cardiomyocyte subsets, and decreased *Top2a* and *Pcna* specifically in the proliferating cardiomyocytes, to support the non-compact phenotype in *Srsf3* iKO hearts.

Our scRNA-seq and bulk RNA-seq provided no evidence of mis-regulated expression of any of the epicardial paracrine signals that have previously been implicated in cardiomyocyte proliferation e.g. *Rspo1*, *Wnt*, retinoic acid, even though these are identified as SRSF3 targets by irCLIP. Despite unchanged levels of transcript per cell, the initial reduction in epicardial coverage would translate into a reduced total level of all of these mitogenic signals and others.

3. The rescue by expansion of non-recombined cells is a most interesting aspect of this study. Can the authors see any such outcompeting in the explant experiments (for example in Figure 2)? Do the authors consider this to be an exclusively *in vivo* competition phenomenon? Given the known roles of *Myc* in cell competition can the authors use their single cell transcriptomic data to score *Myc* expression levels in cells from *Srsf3* iKO hearts or determine if *Myc* transcripts are bound by SRSF3?

The phenomenon we describe (elimination of cells that have undergone cell cycle arrest, apoptosis and senescence, and their complete replacement by the remaining non-targeted cells that continue to proliferate) is not consistent with the classical definition of cell competition, as mediated by *Myc*, which is rather “a tissue homeostasis mechanism by which low-anabolizing but otherwise viable cells are eliminated from tissues due to confrontation with higher-anabolizing cells”. Consistent with this, we do not find evidence of *Myc* up-regulation in the epicardial lineage, as shown in Supplementary Fig. 12E-F by scRNA-seq. Indeed, *Myc* is barely expressed in the embryonic epicardium. Thus, an investigation of cell competition is not warranted on the basis of our data.

We suspect that all epicardial cells (not only those of mutant hearts) possess an intrinsic mechanism to sense epicardial cell density and adjust proliferation to reach a programmed setpoint and complete coverage of the myocardium, via a mechanism that has yet to be determined. We additionally ruled out a mechanism of lateral inhibition by YAP/TAZ.

As elaborated below, we now offer evidence to support the auto-regulation of *Srsf3* splicing as a mechanism to control its own expression levels and drive the compensatory hyper-proliferation (Supplementary Fig. 13).

4. The authors suggest that this rescue occur by upregulation of *Srsf3* in non-recombined cells. It would be helpful to provide additional lines of evidence supporting the hypothesis that SRSF3 expression is upregulated due to hypoxia. Do the CLIP experiments reveal whether SRSF3 binds to its own transcript?

We investigated *Srsf3* splicing in MEC1 epicardial cells. While control-transfected MEC1 included auto-regulatory exon 4 in 10.66% of *Srsf3* transcripts, exon 4 was present in 1.67% (siSrsf3#1) and 3.09% (siSrsf3#2) of transcripts from *Srsf3* knockdown MEC1 cells (Supplementary Fig. 13A). Consistent with the notion of auto-regulation, we detected extensive SRSF3 CLIP-tag clusters across the autoregulatory cassette exon 4 of *Srsf3* (Supplementary Fig. 13B), supporting a mechanism for *Srsf3* auto-up-regulation in surviving cells following *Srsf3* iKO or knockdown. In contrast, the

autoregulatory exons of *Srsf4*, which are known not to be cross-regulated by SRSF3 (Ånkö et al., 2012), were not differentially spliced in *Srsf3* knockdown MEC1 cells (Supplementary Fig. 13A). Collectively, these data support auto-regulation by SRSF3 itself as the predominant mechanism driving elevated expression and hyperproliferation. This mechanism coincides with hypoxia in the iKO myocardium, however whether hypoxia drives auto-regulation of *Srsf3* would entail an extensive investigation that is beyond the scope of the current study.

5. The authors imply that SRSF3 may regulate *Ccnd1* mRNA stability. Can the authors directly evaluate this point? Please clarify if this gene is also affected in the knock-down experiments in MEC1 cells.

We conducted mRNA stability assays to measure *Ccnd1* decay, alongside that of the G1/S regulator *Ccng1*, after inhibiting transcription with Actinomycin D. We found no significant differences in half-life or decay rate constant for either gene after knockdown of *Srsf3* (Supplementary Fig. 9), suggesting that SRSF3 controls cyclin D1 transcription rather than its stability. *Ccnd1* is indeed significantly depleted in MEC1 cells after *Srsf3* knockdown (Volcano plot, Figure 7B, Table 2), as it is in the immortalised epicardial cell line (Figure 8H).

6. Please brighten the immunofluorescence panels in Figure 1 to more clearly show nuclear labelling and tissue structure.

Following a previous round of review, we were asked to reduce immunofluorescence signal intensity of these images to allow distinction of the higher epicardial expression relative to that in cardiomyocytes. We have included higher power insets to Figure 1 to ensure that nuclear labelling can be visualised.

7. Given the broad roles of SRSF3 is the adjective key necessary in the title?

We have removed the word key, even though its use is entirely consistent with the phenotype and with the enrichment of cell cycle-related genes by both irCLIP and RNA-seq. Being a key regulator of proliferation does not preclude additional roles in other processes, but the loss of function phenotype, in vitro validation and compensatory up-regulation argue that *Srsf3* is both sufficient and necessary for epicardial proliferation to underpin formation.

In keeping with the request of the *Development* reviewer (#2) to extend the title to make it more informative, we suggest the title '*The RNA-binding protein SRSF3 controls epicardial formation by regulating splicing and proliferation*' for the new manuscript, but would be guided by the Editor on the final title.

Reviewer #3 (Significance (Required)):

This ms advances understanding of the complex roles of this RNA binding protein in the heart - and raises an important message about how incomplete Cre recombination needs to be considered in interpreting conditional mutant phenotypes. This study would be of interest to researchers in the fields of heart development and RNA-protein interactions. Although there are a number of major points to be addressed, these could be potentially dealt with rapidly.

Development Reviewers comments

Reviewer 1:

Summary:

The study by Irina Lupu and colleagues highlights SRSF3 as a key regulator of epicardial development by regulating epicardial cell proliferation. This was demonstrated via two murine knockout models; the first to assimilate the role SRSF3 plays in epicardial formation as a whole, and the second to address its importance post a pivotal maturation point. Through scRNA sequencing and irCLIP, several SRSF3 targets were ascertained and identified as cell cycle regulators. Those epicardial cells that did not lose SRSF3 compensated the loss of some of their

mates by increasing SRSF3 expression and over-proliferating. Overall, the paper is interesting and the conclusions are largely supported by the provided data.

Major comments:

1. Authors claim that a "reduction in SRSF3 expression levels coincided with the downregulation of WT1 in the epicardium". This was evidenced by immunofluorescence imaging (figure 1A). I suggest conducting a qRT-PCR to quantify *Wt1* expression over time, similar to the experiment they performed in figure 1B.

We did not present *Wt1* mRNA expression in Fig. 1 A for comparison with *Srsf3* for two reasons:

1. Although epicardial *Wt1* declines over the course of development, coronary endothelial cells begin to express *Wt1* from E11.5, increasing thereafter (examples in panel C below). Thus, whole heart qRT-PCR would not show a decline comparable to *Srsf3*.

2. The downregulation of epicardial *Wt1* over the course of development has already been extensively described in our 2020 manuscript (Lupu et al (2020) *Stem Cell Reports*. 14(5):770-787), in which we visualised epicardial *Wt1* by single-molecule RNA in situ hybridization (RNAscope) and quantified expression using a flow cytometry-based RNA ISH method (PrimeFlow RNA Assay).

We now explicitly highlight the comparable decline in *Wt1* and cite this study on page 5 of the revised manuscript. For the reviewer's reference, the relevant figure panels are shown here:

NOTE: Figure provided for reviewer has been removed. It showed selected panels from Figure 4 from Lupu, I.E., Redpath, A.N., Smart, N. (2020) Spatiotemporal Analysis Reveals Overlap of Key Proepicardial Markers in the Developing Murine Heart. *Stem Cell Reports*. 14, 770-787. doi: 10.1016/j.stemcr.2020.04.002. Epub 2020 Apr 30. PMID: 32359445; PMCID: PMC7221110.

A: ISH of E11.5, E13.5, E15.5, and E17.5 hearts to show expression of *Wt1* in the epicardium (n = 3). B) Flow cytometric analysis of *Wt1* in the sorted epicardial population (mean geometric mean fluorescence intensity \pm SEM; n = 4 independent experiments). C) Immunostaining for WT1 and PECAM1 on E11.5 and E12.5 heart sagittal sections reveals expression of WT1 in endothelial cells (ECs).

2. A western blot depicting SRSF3 protein production in controls compared to the knockout model may provide stronger evidence of its depletion (figure 1E).

Whole heart western blot to demonstrate epicardial SRSF3 depletion would be confounded by expression in the more abundant cardiomyocytes. Moreover, we no longer have access to *Srsf3* cKO mice to obtain fresh samples for western blotting. Evidence for *Srsf3* depletion was instead obtained by analysis of *Srsf3* transcripts in single cell RNA-seq, as described under point 3 below.

3. Authors state that they were unable to directly identify the absence of exons 2 and 3 in individual cells. Please provide evidence that exons 2 and 3 have been knocked out, at least by performing a qRT-PCR.

To address this important point, we reanalysed the *Srsf3* iKO single cell RNA-seq data using a customised script that enabled us to demonstrate the recombination and deletion of exon 2/3 in mutant epicardial cells. These data are included in a new Supplementary Figure 4. We maintain that the 3' bias of 10X chromium technology limits the ability to detect recombination in all transcripts, and that this approach is further confounded by the failure of targeted cells to proliferate and to pass standard quality control filters. Thus, the figure of ~10% recombination in mutant epicardial cells is likely severely under-estimated, but importantly validates epicardial deletion with our inducible genetic knockout strategy.

4. To prove the functional implication in the observed phenotype of the identified SRSF3 targets, please interfere with *Map4k4* activity or expression and check whether the defective epicardial cell proliferation is reverted. This should be done at least in vitro, ideally in vivo.

We appreciate the reviewer's comment and the importance of functional validation for implicating SRSF3 targets in epicardial function. Given that proliferation requires coordinated regulation at multiple levels, and SRSF3 is shown by irCLIP to bind hundreds of key cell cycle regulators, restoring expression or splicing of a single gene will likely be insufficient to rescue the extreme reduction in proliferation of Srsf3 knockdown cells. We have made clear that we do not feel this mechanism fully accounts for proliferative arrest in the concluding sentence of that section: *“suggesting that regulation [of Map4k4 splicing] by SRSF3 may, in part, contribute to the switch of epicardial cells from the proliferative to senescent state.”*

Minor comments:

1. Several minor typos and spacing issues were observed. Please correct.

These errors have now been corrected.

2. It would be good for the reader if the authors would simplify their rationale for the use of the two mouse models. It is slightly convoluted and not easy to follow.

We have included a simple explanation and comparison between the two mouse models in the revised manuscript which explicitly states the advantages and limitations of each.

On page 6...*Despite the ST and cardiomyocyte recombination, Tg(Gata5-Cre) enables efficient targeting of E9.5 proepicardial progenitors, that cannot be achieved with inducible epicardial Cre lines.*

On page 8 (in relation to the Wt1CreERT2 line)...*The restricted window for induction prioritises specificity over efficiency. While cEC and cardiomyocyte targeting can be largely avoided with this strategy, it produces mosaic recombination, even with our optimised tamoxifen regime. Thus, while both constitutive and inducible epicardial Cre lines are imperfect, they are considered the best available, and their combined use partly mitigates their respective inadequacies.*

3. In figure 4, it is recommended to add a stacked bar plot to represent the percentage of each cell cluster/population after 4A. This would help the reader

We now include a stacked bar plot (Figure 4B) to represent the percentage of each cell cluster.

4. Figure 4B. It is confusing for the reader to understand the fact that the majority of tdTomato+ sorted cells in Srsf3 iKO keep expressing Srsf3. Including the quantification of the image could help.

Indeed, the retention of SRSF3 in the majority of tdTomato+ sorted cells seems counter-intuitive. Despite the widespread reliance upon Cre reporter lines, the highly efficient recombination of the short transcriptional stop sequence of Cre reporters does not accurately reflect recombination across multiple exons for gene deletion, which is typically much less efficient, as widely reported (Fernandez-Chacon et al., 2019; Payne et al., 2018; Redpath and Smart, 2021). Moreover, as we show, Srsf3-depleted cells are compromised by impaired proliferation and viability. Some are lost in vivo, others likely do not adhere after sorting, and the enrichment of the Srsf3-retaining proliferative population would be further exaggerated after 48 hours in culture.

Reviewer 2:

In this study Lupu and colleagues convincingly demonstrate that SRSF3 as an important regulator of cell cycle in the epicardium. This is shown by two series of conditional deletion experiments and molecular analyses and the work is well documented and clearly presented. The proliferation role appears to be consistent with the role of SRSF3 in other cell types. In addition the authors observed epicardial recovery by unrecombined cells and make an important general point about mosaicism and phenotyping in conditional mutagenesis experiments. The authors have discussed in detail how they plan to address an earlier set of comments. Their replies are well argued and fair, and the proposed revisions include useful experiments such as identifying splicing changes in mutant hearts

using bulk RNA sequencing and investigating the origin of *Srsf3* expressing cells in mosaic hearts. I have only a small number of additional points.

1. Can the authors rule out an indirect role in myocardial proliferation that might contribute to the compaction defect using the later activated Cre line? The authors could expand the last paragraph of the discussion to compare the new data with the known role of SRSF3 in regulating proliferation in other contexts.

We acknowledge that the delay in formation of an intact epicardial layer would severely diminish the amount of mitogenic signalling required for myocardial proliferation and compaction, as would failed coronary vasculature formation. These are both recognised within the manuscript as having important indirect roles in myocardial proliferation and compaction. As the reviewer will appreciate, distinguishing between the epicardial requirements for paracrine signalling, direct cell contributions and directing coronary vessel expansion, which all contribute to the observed compaction defect, is not trivial. We feel the contribution of each of these mechanisms has been suitably described in our manuscript.

2. Concerning *Myc* and cell competition, the authors demonstrate that *Myc* transcripts are bound by SRSF3 (page 11), suggesting that transcript processing may be impacted. They should discuss the possibility that this might place cells with normal *Myc* processing at an advantage.

The phenomenon we describe (elimination of cells that have undergone cell cycle arrest, apoptosis and senescence, and their complete replacement by the remaining non-targeted cells that continue to proliferate) is not consistent with the classical definition of cell competition, as mediated by *Myc*, which is rather “a tissue homeostasis mechanism by which low-anabolizing but otherwise viable cells are eliminated from tissues due to confrontation with higher-anabolizing cells”. Consistent with this, we do not find evidence of *Myc* up-regulation in the epicardial lineage, as shown in Supplementary Fig. 12E-F by scRNA-seq.

Although SRSF3 binding to the *Myc* transcript was identified in the immortalised epicardial cell line, *Myc* is barely expressed *in vivo* in the embryonic epicardium (Supplementary Fig. 12E-F scRNA-seq), nor was its expression altered in the MEC1 cell line after *Srsf3* knockdown (Table 2). The lack of evidence to support MYC-mediated cell competition is discussed in the revised manuscript. A more detailed investigation of *Myc* processing is not warranted on the evidence of our data. Instead, we offer evidence to support the auto-regulation of *Srsf3* splicing as a mechanism to control its own expression levels and promote compensatory hyper-proliferation (Supplementary Fig. 13).

3. The title as it stands, with or without the word “key”, is not very informative. Perhaps the authors can extend it, for example mentioning epicardial proliferation.

We thank the reviewer for this valuable comment. Accordingly, we suggest changing the title to ‘The RNA-binding protein SRSF3 controls epicardial formation by regulating splicing and proliferation’. We are happy to be guided by the Editor on the final title.

Second decision letter

MS ID#: dev.204918R1

MS Title: The RNA-binding protein SRSF3 controls epicardial formation by regulating splicing and proliferation

Authors: Irina-Elena Lupu; Susann Bruche; Anob M. Chakrabarti; Ian R. McCracken; Quang M. Dang; Tamara Carsana; Sarah De Val; Andia Redpath; Nicola Smart

Dear Dr Smart,

I am happy to tell you that your manuscript has been accepted for publication in Development, pending our standard publication integrity checks.

Reviewer 1

The authors have appropriately addressed the issues raised in the revision. I congratulate the authors for this very interesting paper.

Reviewer 2

SUMMARY OF THE ADVANCE MADE IN THIS PAPER AND ITS POTENTIAL SIGNIFICANCE TO THE FIELD

This data-rich manuscript provides new mechanistic insights into epicardial and cardiac development as well as raising an important issue about phenotypic compensation due to incomplete Cre recombination.

SUGGESTIONS TO AUTHORS

The authors have very thoroughly addressed earlier comments by adding new experiments and discussion and extending the mechanistic insights of this study.